

# Estuarine microbial networks and relationships vary between environmentally distinct communities

Sean R. Anderson[1,2] and  Elizabeth L. Harvey[3]

[1] Northern Gulf Institute, Mississippi State University, Mississippi State, MS, United States of America
[2] Ocean Chemistry and Ecosystems Division, Atlantic Oceanographic and Meteorological Laboratory, National Oceanic and Atmospheric Administration, Miami, FL, United States of America
[3] Department of Biological Sciences, University of New Hampshire, Durham, NH, United States of America

## ABSTRACT

Microbial interactions have profound impacts on biodiversity, biogeochemistry, and ecosystem functioning, and yet, they remain poorly understood in the ocean and with respect to changing environmental conditions. We applied hierarchical clustering of an annual 16S and 18S amplicon dataset in the Skidaway River Estuary, which revealed two similar clusters for prokaryotes (Bacteria and Archaea) and protists: Cluster 1 (March-May and November-February) and Cluster 2 (June-October). We constructed co-occurrence networks from each cluster to explore how microbial networks and relationships vary between environmentally distinct periods in the estuary. Cluster 1 communities were exposed to significantly lower temperature, sunlight, $NO_3$, and $SiO_4$; only $NH_4$ was higher at this time. Several network properties (*e.g.*, edge number, degree, and centrality) were elevated for networks constructed with Cluster 1 vs. 2 samples. There was also evidence that microbial nodes in Cluster 1 were more connected (*e.g.*, higher edge density and lower path length) compared to Cluster 2, though opposite trends were observed when networks considered Prokaryote-Protist edges only. The number of Prokaryote-Prokaryote and Prokaryote-Protist edges increased by >100% in the Cluster 1 network, mainly involving Flavobacteriales, Rhodobacterales, Peridiniales, and Cryptomonadales associated with each other and other microbial groups (*e.g.*, SAR11, Bacillariophyta, and Strombidiida). Several Protist-Protist associations, including Bacillariophyta correlated with Syndiniales (Dino-Groups I and II) and an Unassigned Dinophyceae group, were more prevalent in Cluster 2. Based on the type and sign of associations that increased in Cluster 1, our findings indicate that mutualistic, competitive, or predatory relationships may have been more representative among microbes when conditions were less favorable in the estuary; however, such relationships require further exploration and validation in the field and lab. Coastal networks may also be driven by shifts in the abundance of certain taxonomic or functional groups. Sustained monitoring of microbial communities over environmental gradients, both spatial and temporal, is critical to predict microbial dynamics and biogeochemistry in future marine ecosystems.

Corresponding authors
Sean R. Anderson,
sean.r.anderson@noaa.gov
Elizabeth L. Harvey,
elizabeth.harvey@unh.edu

## INTRODUCTION

Over the last two decades, the expansion of high-throughput environmental sequencing, or amplicon metabarcoding, has improved our ability to monitor Bacteria, Archaea, and protists in the ocean (*Sogin et al., 2006*; *Xia, Guo & Liu, 2017*; *Caron & Hu, 2019*; *Santoferrara et al., 2020*; *Burki, Sandin & Jamy, 2021*). Amplicon surveys have informed marine microbial dynamics, revealing spatial trends in biodiversity (*Ibarbalz et al., 2019*), recurrent seasonal patterns in composition (*Ward et al., 2017*; *Chafee et al., 2018*; *Giner et al., 2019*), and more accurate representation of rare or cryptic organisms, especially among protists (*Burki, Sandin & Jamy, 2021*). These monitoring efforts have also reinforced the importance of environmental and biological factors in structuring microbial communities (*Fuhrman, Cram & Needham, 2015*; *Logares et al., 2020*). For instance, several environmental factors, like temperature, nutrients, sunlight, and water depth, consistently influence microbes on regional to global scales (*Gilbert et al., 2012*; *Sunagawa et al., 2015*). Microbes are also driven by interactions that they have with each other, with such relationships being subject to environmental stressors (*Piccardi, Vessman & Mitri, 2019*; *Hernandez et al., 2021*).

Microbial interactions underpin ocean ecosystem functioning, influencing carbon transfer, nutrient cycling, and organic matter transformation (*Azam et al., 1983*; *Worden et al., 2015*). Microbes are highly interconnected, exhibiting a range of interactions, from parasitism and predation to symbiosis and mutualism, all differentially influencing community diversity and biogeochemical cycles (*Worden et al., 2015*). Despite their importance, many interactions remain unresolved. A recent literature review of ~2500 microbial interactions found that 14% were ambiguous across aquatic environments (*Bjorbækmo et al., 2020*). Realistically, the number of unresolved interactions in nature is likely much higher, owing to the large functional diversity of microbes, the challenges in culturing them in the lab, and the lack of tools available to examine such interactions at scale and under high taxonomic resolution (*Krabberød et al., 2017*). Given the important role of microbial interactions in shaping biodiversity and ecosystem functioning, it remains critical to better resolve these interactions and how they may shift over time and space and under different environmental conditions.

Amplicon surveys coupled with co-occurrence network analysis represent a powerful method to infer microbial relationships, establishing significant correlations between amplicon sequence variants (ASVs) based on relative abundance data (*Röttjers & Faust, 2018*; *Faust, 2021*). Co-occurrence networks have been conducted across marine systems, revealing globally important microbial associations that may reflect interactions, like parasitism or symbiosis (*Lima-Mendez et al., 2015*), and providing context for phytoplankton bloom dynamics, species succession, and biogeochemistry (*Needham & Fuhrman, 2016*; *Needham, Sachdeva & Fuhrman, 2017*; *Bolaños et al., 2021*). More recently, a reanalysis of Tara Ocean data revealed that global networks were structured by environmental (poleward) niches and influenced by factors like temperature, salinity, and nutrients (*Chaffron et al., 2021*). Together, these studies emphasize the importance

of microbial networks to ecosystem functioning and food web dynamics, as well as their potential susceptibility to environmental changes.

A universal challenge in constructing microbial networks is how to deal with environmental factors that are known to strongly influence communities (*Faust, 2021*). The most common strategy to assess environmental impact has been to aggregate samples into a single network, including environmental variables as additional nodes that can be correlated to ASVs (*Fuhrman, Cram & Needham, 2015*; *Needham, Sachdeva & Fuhrman, 2017*). However, this approach often results in fewer environmental network correlations compared to those between species (*Gilbert et al., 2012*; *Chow et al., 2014*). Another strategy involves separating ASV tables into groups, for instance based on temporal (seasons or years) or spatial scales (water depth), and comparing resulting networks (*Lima-Mendez et al., 2015*; *Milici et al., 2016*; *Kellogg et al., 2019*; *Lambert et al., 2021*). Networks can also be separated by binning nodes (*Röttjers & Faust, 2018*; *Chaffron et al., 2021*) but tools are limited (*Faust, 2021*) and have not been well applied to marine samples.

Another potential method is to separate samples for network analysis *a priori* based on community composition (beta diversity). Separating samples in this manner may be relevant for estuarine or other coastal microbial communities that are seasonally structured and sensitive to anthropogenic impacts (*Fuhrman et al., 2006*; *Chafee et al., 2018*). Many estuaries along the southeastern U.S., including the Skidaway River Estuary (GA, U.S.), were once considered pristine and are now threatened by habitat transformation and nutrient loading (*Verity, Alber & Bricker, 2006*). Though well mixed from semidiurnal tides, long-term monitoring in the Skidaway River has revealed increased nutrient input correlated to increased abundance of heterotrophic bacteria and most plankton groups (*Verity & Borkman, 2010*). Continued anthropogenic input of nutrients in this region (and others) may enhance warming, eutrophication, and habitat loss, with implications for fisheries and ecosystem management (*Verity, Alber & Bricker, 2006*). Identifying and characterizing microbial relationships that occur over environmentally distinct periods may provide insight into their long-term dynamics in changing coastal habitats.

The aim of our study was to assess how different environmental conditions influence relationships among microbes (Bacteria, Archaea, and protists) in the Skidaway River Estuary. We performed 16S and 18S amplicon surveys on a weekly-monthly basis (33 days) and constructed microbial networks with co-occurrence analysis. Instead of binning samples categorically, we separated ASV tables for network analysis based on hierarchical clustering of beta diversity. This approach revealed two distinct communities, similar for prokaryotes and protists, representative of contrasting environmental conditions, mainly temperature and nutrients. Despite the same initial number of ASVs used in each network, we observed differences in network properties (*e.g.*, degree, centrality, and number of edges) between clusters, as well as changes in the types of associations that occurred. The response of microbial relationships to anthropogenic threats and changing conditions remains unclear (*Caron & Hutchins, 2013*), and so, it is critical to better understand how current networks vary between environmentally distinct periods.

## MATERIALS & METHODS

### Sample collection

Surface water samples (1 m) were collected weekly to monthly from March 2017 to February 2018 in the Skidaway River Estuary (latitude, 31°59′25.7′N; longitude, 81°01′19.7′W), encompassing 33 sampling days. For consistency between weeks, sampling always occurred at high tide. Water samples were collected with a 5-L Niskin bottle, filtered on site through 200-μm mesh (to exclude zooplankton) into a 20-L carboy, and transferred to a nearby lab for processing. Samples (250–1000 ml) were filtered in triplicate from the 20-L carboy through 47-mm, 0.22-μm polycarbonate filters (Millipore) using vacuum filtration. Filters were stored at −80 °C. On three days (8/30, 10/11, and 11/21), only two biological replicates were filtered.

Surface temperature, salinity, and dissolved oxygen were measured using a YSI (600S sonde). Solar radiation data was collected from a nearby land-based site on Skidaway Island. Triplicate chlorophyll samples (50–100 ml) were filtered onto 0.7-μm GF/F filters, extracted in 91% ethanol, and measured on a Turner AU10 fluorometer (*Graff & Rynearson, 2011*). Dissolved nutrients ($NO_3$, $NH_4$, $PO_4$, and $SiO_4$) were measured *via* a Technicon AutoAnalyzer (SEAL Analytical), while particulate organic carbon (POC) and nitrogen (PON) were measured using a Thermo Flash elemental analyzer (*Bittar et al., 2016*; *Anderson & Harvey, 2019*). Dissolved nutrients and POC/PON were not measured on 9/6; these samples were not considered in the constrained ordination.

### PCR conditions and DNA sequencing

The DNeasy PowerSoil kit (Qiagen) was used to extract DNA following manufacturer's protocols. DNA samples were eluted in 10 mM Tris–HCl (pH = 8.5). DNA concentrations were estimated with the Qubit dsDNA HS kit (Thermo Scientific) and ranged from 2–5 ng μl$^{-1}$ per sample. A two-step PCR approach was employed with two different primer sets, targeting prokaryotes (16S rRNA) or protists (18S rRNA). We used the following primers to target the V4 region of the 18S rRNA gene (*Stoeck et al., 2010*): forward (5′-CCAGCASCYGCGGTAATTCC-3 ′) and reverse (5′-ACTTTCGTTCTTGATYRA-3′). For 16S, primers targeted the V4–V5 region (*Parada, Needham & Fuhrman, 2016*): forward (5′-GTGYCAGCMGCCGCGGTAA-3′) and reverse (5′-CCGYCAATTYMTTTRAGTTT-3′). Illumina adapters were attached to each target-specific primer region. 18S PCR conditions involved an initial denaturation step at 98 °C for 2 min, 10 cycles of 98 °C for 10 s, 53 °C for 30 s, and 72 °C for 30 s, followed by 15 cycles of 98 °C for 10 s, 48 °C for 30 s, and 72 °C for 30 s, and a final extension of 72 °C for 2 min (*Stoeck et al., 2010*; *Hu et al., 2015*). 16S PCR conditions consisted of an initial denaturation of 95 °C for 2 min, 25 cycles of 95 °C for 45 s, 50 °C for 45 s, and 68 °C for 90 s, followed by a final elongation step of 68 °C for 5 min (*Parada, Needham & Fuhrman, 2016*). PCR products were purified and size-selected using AMPure XP Beads (A63881; Beckman Coulter). A second PCR step was carried out by attaching dual Illumina indices (P5 and P7) and adapters to template DNA using the Nextera XT Index Kit. Two separate sequencing runs were performed using an Illumina MiSeq (2 ×250 bp for 18S; 2 ×300 bp for 16S) at the Georgia Genomics and Bioinformatics Core at the University of Georgia.

## Bioinformatics

Demultiplexed 16S and 18S FASTQ files were imported and processed separately in QIIME 2 (*Bolyen et al., 2019*). Amplicon sequence variants (ASVs) were inferred with paired-end DADA2 (*Callahan et al., 2016*). Truncation lengths of the forward and reverse reads were defined based on read-quality profiles; otherwise, default DADA2 parameters were used. Protist taxonomy was inferred using QIIME 2-compatible files from the Protist Ribosomal Reference (PR2) database (Version 4.12.0; *Guillou et al., 2013*), while prokaryotic taxonomy was assigned using the SILVA database (Version 138; *Pruesse et al., 2007*). For both gene regions, a Naïve Bayes Classifier was used to train the sequences against reference databases using the feature-classifier plugin in QIIME 2 (*Bokulich et al., 2018*). QIIME 2 taxonomy and count table artifact files (.qza files) were imported into R (Version 3.6.3; *R Core Team, 2020*) using the read_qza function from the qiime2R package (https://github.com/jbisanz/qiime2R).

Sequences were deposited at the Sequence Read Archive of the National Center for Biotechnology Information (NCBI) and made publicly available under accession numbers PRJNA575563 (18S) and PRJNA680039 (16S). R code used for data analysis, including a full list of R packages, is on GitHub (https://github.com/sra34/SkIO-network). This project has been archived on Zenodo (https://doi.org/10.5281/zenodo.6549350).

## Statistical analyses

Community dynamics were investigated separately for 16S and 18S datasets in R, using packages including phyloseq (*McMurdie & Holmes, 2013*), vegan (*Oksanen et al., 2018*), and tidyverse (*Wickham et al., 2019*). Average values presented in the text refer to the mean. To focus on protists, we removed eukaryotes within Metazoa and Streptophyta that were amplified with the 18S primers. Unassigned reads at the supergroup level for protists (PR2 Rank 2) or domain level for prokaryotes (SILVA Rank 1) were also removed, as well as prokaryotic reads assigned to chloroplasts or mitochondria. After filtering, there were 51,770 sequence reads on average across samples for protists (16,857–84,369), assigned to 8,700 ASVs. There were 81,165 sequences on average for prokaryotes (47,250–177,521), assigned to 15,716 ASVs. Two samples in the 16S dataset were removed (3/16 B and 9/20 C) due to low sequence read numbers (94 total samples for 16S *vs.* 96 for 18S). Rarefaction curves were generated using the R package ranacapa (*Kandlikar et al., 2018*). Unfiltered taxonomic assignments and read counts for microbial ASVs are provided in Table S1.

To assess community dynamics, singletons were removed (except for alpha diversity) and samples were rarefied to the minimum read count for 16S (47,249) and 18S tables (16,849), respectively (*Weiss et al., 2017*). Community composition was correlated to environmental variables with distance-based redundancy analysis (dbRDA; *Oksanen et al., 2018*). Constrained ordinations were run with unweighted UniFrac distance matrices and log-transformed environmental factors. Environmental variables that were significant to the ordination (ANOVA, $p$-values <0.05) were identified using the ordistep function (both directions) in vegan with 999 permutations (*Oksanen et al., 2018*). After a final dbRDA run, significant variables were added to the ordination as arrows. Hierarchical clustering (Ward's method) of UniFrac distances was performed using the hclust function
in vegan. The optimal number of clusters was evaluated based on average silhouette widths, a measure of the similarity between each sample and its cluster compared to its similarity to other clusters (*Rousseeuw, 1987*).

Group-specific 16S and 18S relative abundance was assessed over the year and local regression (loess) curves were applied to visualize temporal trends using the geom_smooth function in ggplot2 (*Wickham et al., 2019*). Relative abundances were correlated with environmental variables using Spearman rank correlations ($R$), considering only variables that were significant to the dbRDA. We focused on the most relatively abundant prokaryotes and protists in the dataset at the order level (>2% on average). Observed richness and Shannon diversity were estimated with the estimate_richness function in phyloseq (*McMurdie & Holmes, 2013*). Singletons were considered for diversity estimates to account for rare microbes. Shapiro–Wilks's normality tests were applied to the diversity data, whereafter either paired t-tests (richness) or Wilcoxon tests (Shannon) were used to compare mean diversity or richness between clusters. Similar comparative tests (Wilcoxon or $t$-test) were performed for environmental variables, after checking how each variable was distributed (Shapiro–Wilks).

## Covariance networks

Microbial association networks were constructed for separate clusters using the SParse Inverse Covariance estimation for Ecological Association and Statistical Inference (SPIEC-EASI; Version 1.1.0) package in R (*Kurtz et al., 2015*). SPIEC-EASI uses ASV count tables as input and computes an inverse covariance matrix, using conditional independence to infer direct associations (*Kurtz et al., 2015*; *Röttjers & Faust, 2018*). The program also supports merging ASV tables across gene marker regions, an approach that has been tested previously to investigate cross-domain associations (*Tipton et al., 2018*). SPIEC-EASI aims to be robust to the compositional nature of amplicon data and aims to infer sparse networks that are more conservative against false-positive or indirect edges (*Kurtz et al., 2015*).

Networks constructed with too many edges can result in "hairball" networks that are difficult to interpret and may yield ambiguous relationships (*Röttjers & Faust, 2018*). Therefore, ASV tables were filtered for network analysis to include the top 150 most abundant (based on sequence reads) 16S and 18S ASVs (300 total ASVs) per cluster. Covariance networks were constructed with the spiec.easi function using ASV count tables as input (with matching sample IDs) and the "mb" (Meinshausen–Buhlmann) neighborhood selection setting. Two samples that were excluded from the 16S dataset due to low read numbers (3/16 B and 9/20 C) were also filtered from the 18S set at this stage to support merging of ASV tables. The Stability Approach to Regularization Selection (StARS) was used to select the optimal sparsity parameter with a threshold set to 0.05 (*Liu, Roeder & Wasserman, 2010*). The spiec.easi function performs centered log-ratio (clr) transformation of ASV count tables, eliminating the need to pre-transform abundance data (*Tipton et al., 2018*). SPIEC-EASI outputs a correlation matrix of positive and negative values (weights) for all significantly paired edges.

Networks were visualized in Cytoscape (*Shannon et al., 2003*). The total number of positive and negative edges were compared between networks for domain level associations

(*e.g.*, Protist-Protist, Prokaryote-Protist, and Prokaryote-Prokaryote), as well as the most prevalent types of order level (SILVA Rank 4; PR2 Rank 5) relationships within each broader category. Topological features were measured using the NetworkAnalyzer plugin (*Assenov et al., 2008*), analyzing networks overall or based on the major domain level pairings. Edge density and average path length were measured for each network, which in this case, referred to the fraction of realized to potential microbial edges and the average distance between any two microbial nodes (*Faust & Raes, 2012*). Degree and closeness centrality were estimated for each node (or ASV). Degree refers to the number of edges connected to a given microbe (represented as nodes), whereas closeness centrality indicates the proximity of a given microbe to all other microbes in the network; higher centrality indicates a greater contribution to network connectivity (*Röttjers & Faust, 2018*). Centrality and degree data were non-normal (Shapiro–Wilks) and compared at the domain level and for the most relatively abundant class level groups (SILVA Rank 3; PR2 Rank 4) over the year (Wilcoxon tests).

## RESULTS

Several environmental variables fluctuated over the year (Table S2), most notably temperature, $SiO_4$, and $NO_3$, all peaking in June-October. Temperature and $SiO_4$ were strong covariates in the dataset (Spearman $R = 0.86$, $p$-value <0.001). Temperature also covaried with $NO_3$ and $PO_4$ (both with Spearman $R = 0.48$, $p$-values <0.01), as well as $NH_4$ (Spearman $R = -0.73$, $p$-value <0.001). Other factors like $PO_4$, salinity, POC, and PON were less variable, while $NH_4$ peaked in December-February (Table S2). Chlorophyll (<200 $\mu$m) ranged from 1.32–6.39 $\mu$g $L^{-1}$, varying more greatly between sampling intervals from March-August (Table S2).

### Environmental impact on prokaryotes and protists

For both prokaryotes and protists, the number of read counts *vs.* ASVs was saturated across samples (Fig. S1). Several dominant prokaryotic and protist groups were temporally variable and significantly correlated with environmental variables, particularly temperature, $NO_3$, and $SiO_4$ (Figs. 1; 2). Temporally variable prokaryotes included Actinomarinales, which peaked in abundance in June-October and was strongly correlated to temperature (Spearman $R = 0.8$, $p$-value <0.001), as well as Flavobacteriales, Rhodobacterales, and Oceanospirillales that contributed most to relative abundance in March-May or November-February and were negatively correlated to temperature (Spearman $R = -0.59$ to $-0.8$, $p$-values <0.05; Figs. 1A–1B). SAR11 Clade, the most abundant prokaryotic group on average over the year, became most relatively abundant in November-February (Fig. 1A); however, SAR11 abundance was not significantly correlated to any environmental factor tested (Fig. 1B).

Several protist groups, like Unassigned Dinophyceae and Dino-Groups I and II (Syndiniales), were most relatively abundant in June-October and positively correlated to temperature (Spearman $R = 0.68 - 0.83$, $p$-values <0.001; Figs. 2A–2B). Other groups, like Peridiniales, Gymnodiniales, and Cryptomonadales, peaked in either March-May or November-February and were negatively correlated to temperature (Spearman $R = -0.42$

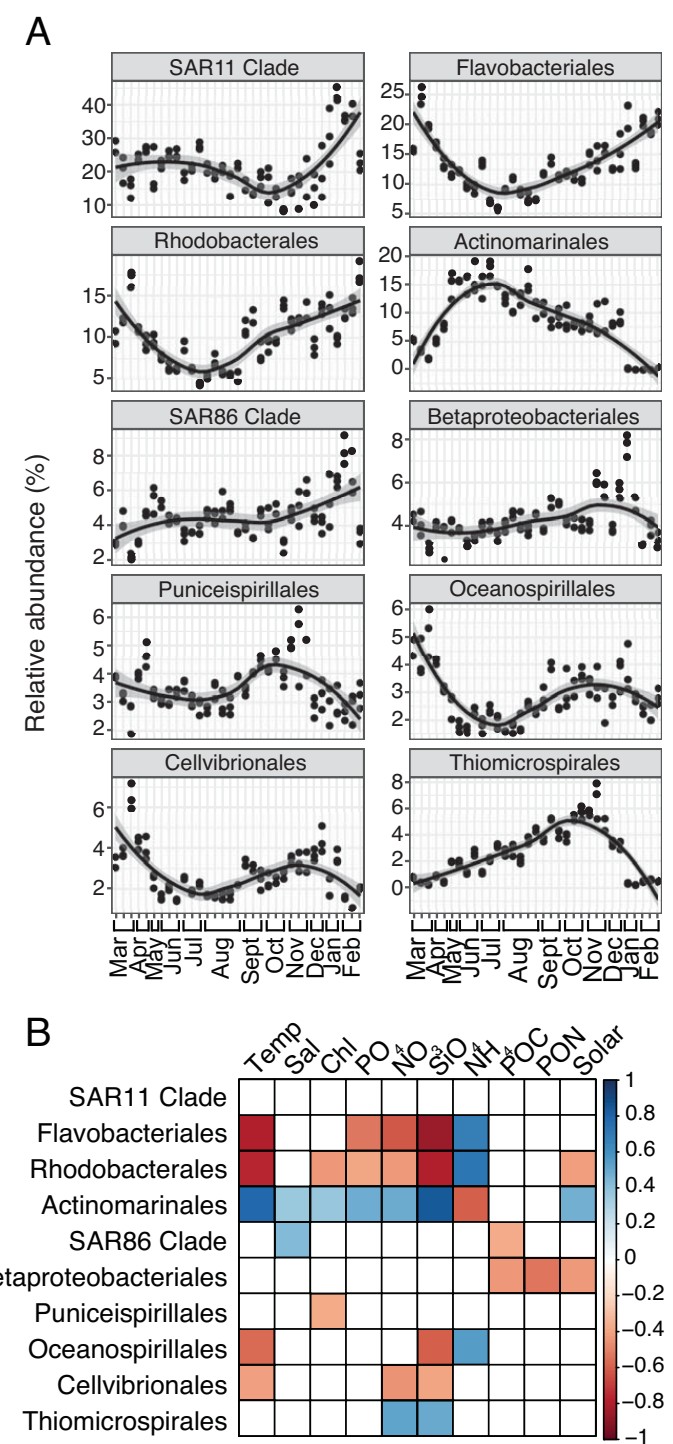

**Figure 1** **Group-specific 16S relative abundance over the year and correlation to environmental variables.** (A) Relative abundance (%) of the top ten most relatively abundant order level prokaryotes over the year (> 2% relative abundance). Local regression (loess) curves (continued on next page...)

**Figure 1 (…continued)**
represent smoothed trends (black lines) and 95% confidence intervals (shaded gray). Abundance data is
shown in triplicate or duplicate (8/30, 10/11, and 11/21) and samples are grouped by month on the $x$-axis.
(B) Spearman correlations between group-specific prokaryotic relative abundance and environmental
variables. Only significant correlations are shown (Spearman $R$, $p$-values $< 0.05$), with the sign of corre-
lation indicated by a red (negative) to blue (positive) color gradient. Stronger correlations are represented
by darker colors. White boxes indicate no significant correlation.

to $-0.56$, $p$-values $< 0.05$; Figs. 2A–2B). Bacillariophyta and Mamiellales were the most
abundant protist groups on average over the year (Fig. 2A). Bacillariophyta relative
abundance was lowest in March-May and positively correlated with $SiO_4$ (Spearman
$R = 0.36$, $p$-values $< 0.05$; Figs. 2A–2B). Group-specific abundances of Mamiellales,
Strombidiida, and Choreotrichida were consistent and not correlated with any factors
(Figs. 2A–2B). Microbial groups that were positively (or negatively) correlated with
temperature were inversely correlated with $NH_4$ (Figs. 1B; 2B).

### Hierarchical clustering to distinguish separate communities

Two clusters were determined to be optimal for each dataset based on plots of average
silhouette widths, here referred to as Clusters 1 and 2 (Fig. S2). Hierarchical clustering
revealed a more separated cluster of samples from March-May and November-February
(Cluster 1) and a tightly grouped set of samples from June-October (Cluster 2; Figs.
S3–S4). 16S and 18S samples were clustered in the same manner (Fig. 3). Distance-based
redundancy analysis (dbRDA) revealed temporal variability in microbial composition, with
significant environmental variables explaining 58% (18S) and 63% (16S) of the variance
from the sum of the first two axes (Figs. 3A–3B). Temperature was among the strongest
variables significantly influencing the ordination (ANOVA, $p$-value $= 0.01$), distinguishing
Cluster 2 samples from Cluster 1 (Fig. 3). Other factors like $SiO_4$ and $NO_3$ were also
significant constraints on the dbRDA (ANOVA, $p$-values $< 0.01$) and distinguished Cluster
2 samples (Fig. 3). In contrast, lower $NH_4$ was a significant explanatory factor (ANOVA,
$p$-value $= 0.01$) for the change in composition from Cluster 1 to 2 samples (Fig. 3).

Cluster 1 and 2 microbial communities experienced different environmental conditions
(Table 1). For example, temperature, sunlight, and nutrients (except for $NH_4$) were
significantly higher (Wilcoxon or paired $t$-test, $p$-values $< 0.05$) in Cluster 2 $vs.$ 1 (Table
1). Mean observed protist richness and Shannon diversity were not significantly different
between clusters (Wilcoxon or paired $t$-test, $p$-values $> 0.2$; Table 1), though for prokaryotes,
both richness and diversity were significantly higher (Wilcoxon or paired $t$-test, $p$-values
$< 0.01$) in Cluster 2 compared to 1 (Table 1).

### Network analysis of sample clusters

SPIEC-EASI was used to identify significant statistical correlations between ASVs (based
on their abundance), with such associations (or edges) indicating potential relationships
between microbes. These putative relationships may or may not reflect true biological
interactions. Cluster 1 samples formed a network with 1086 edges (66% positive) between
300 ASVs, while the Cluster 2 network revealed 707 edges (59% positive) between 297
ASVs (Table 2; Fig. S5). Three ASVs were not connected to any other ASV in the Cluster

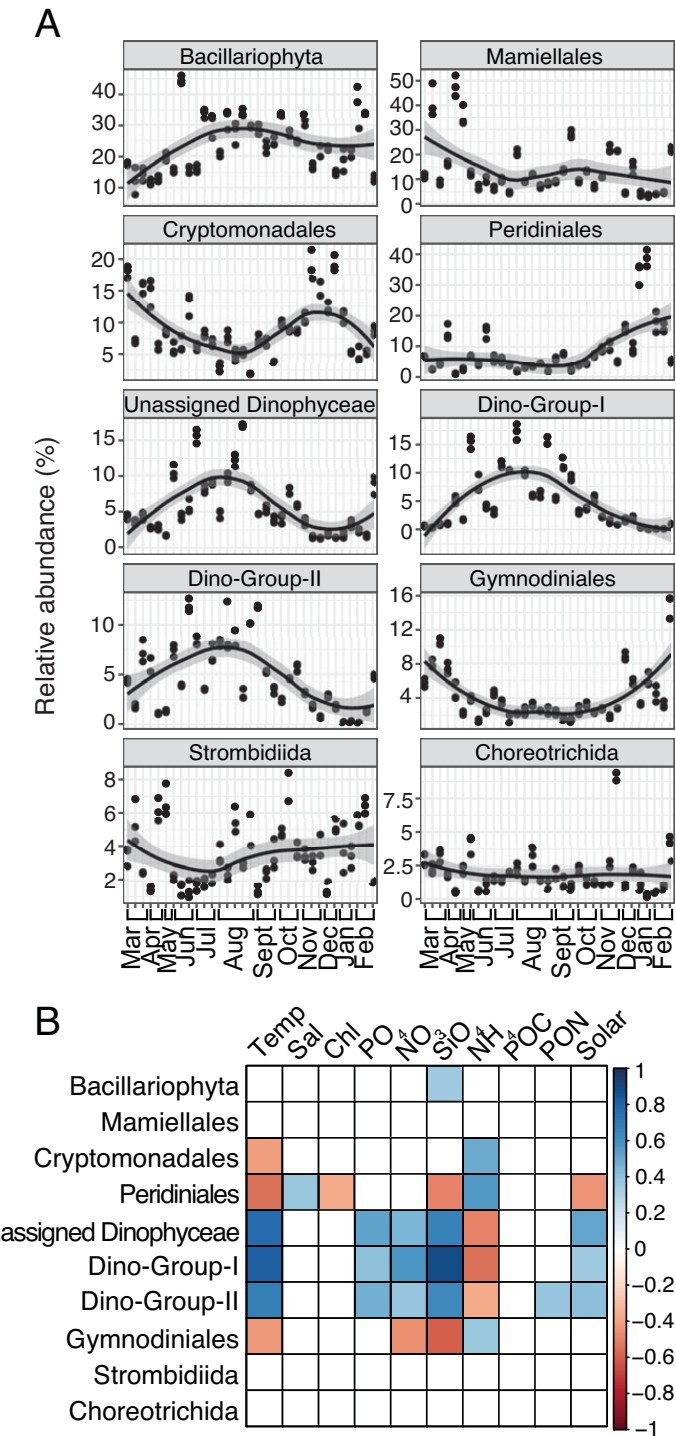

**Figure 2  Group-specific 18S relative abundance over the year and correlation to environmental variables.** (A) Relative abundance (%) of the top ten most relatively abundant order level protists. (B) Spearman correlations (Spearman *R*, *p*-values < 0.05) between group-specific protist relative abundance and environmental variables. Other details are identical to Fig. 1.

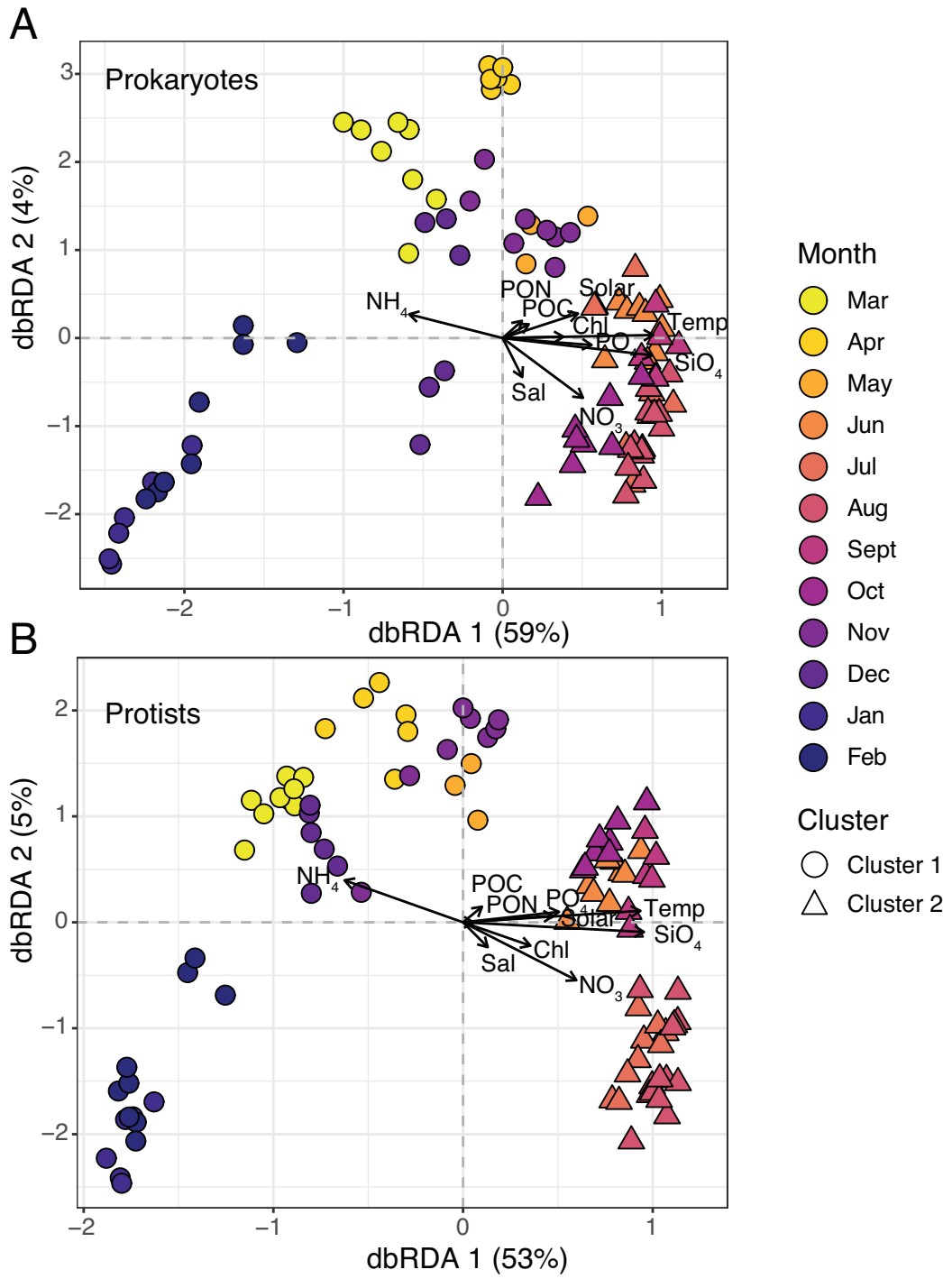

**Figure 3** **Microbial beta diversity in the estuary.** Ordination *via* distance-based redundancy analysis (dbRDA) for prokaryotes (A) and protists (B). Samples are shown in triplicate (or duplicate) and colored by month using a viridis color gradient. Environmental factors are represented as biplot arrows. Temp = temperature (°C); $SiO_4$ = silicate ($\mu$M); $NO_3$ = nitrate ($\mu$M); $PO_4$ = phosphate ($\mu$M); $NH_4$ = ammonium ($\mu$M); Chl = chlorophyll ($\mu$g L$^{-1}$); Sal = salinity (psu); Solar = solar radiation (MJ m$^{-2}$); POC/PON = particulate organic carbon or nitrogen ($\mu$g C or N L$^{-1}$). Sample shapes indicate separate clusters identified *via* hierarchical clustering (Cluster 1 = circles; Cluster 2 = triangles).

**Table 1  Differences in variables between clusters.** Mean and standard deviation of environmental variables, species richness, and Shannon diversity between Cluster 1 and Cluster 2 samples. Prokaryotes (16S) and protists (18S) were clustered similarly ($n = 16$ for Cluster 1; $n = 17$ for Cluster 2). Replicate samples were considered for diversity metrics and varied between 16S ($n = 46$ and 48 for Cluster 1 and 2) and 18S ($n = 47$ and 49 for Cluster 1 and 2) due to removal of two 16S samples (3/16 B and 9/20 C) with low sequence read numbers. Means were compared between clusters, with significantly different variables indicated by an asterisk (Wilcoxon or paired t-tests, * $p$-value $< 0.05$; ** $p$-value $< 0.01$). Temp = temperature (°C); $SiO_4$ = silicate (μM); $NO_3$ = nitrate (μM); $PO_4$ = phosphate (μM); $NH_4$ = ammonium (μM); Chl = chlorophyll (μg $L^{-1}$); Sal = salinity (psu); Solar = solar radiation (MJ $m^{-2}$); POC/PON = particulate organic carbon or nitrogen (μg C or N $L^{-1}$).

| Variables | Cluster 1 | Cluster 2 |
|---|---|---|
| Temp** | 16.43 (5.04) | 28.39 (1.99) |
| $SiO_4$** | 44.83 (13.97) | 105.79 (20.90) |
| $NO_3$** | 0.44 (0.29) | 0.98 (0.46) |
| $PO_4$** | 0.56 (0.12) | 0.67 (0.11) |
| $NH_4$** | 2.4 (1.4) | 1 (0.93) |
| Chl | 3.15 (1.26) | 3.79 (1.43) |
| Sal | 29.22 (1.44) | 29.36 (1.55) |
| Solar* | 9.57 (5.69) | 13.83 (3.66) |
| POC | 60.27 (30.12) | 62.83 (14.67) |
| PON | 10.8 (4.41) | 11.41 (3.4) |
| 18S richness | 488.15 (124.71) | 500.78 (77.22) |
| 18S Shannon | 4.3 (0.48) | 4.47 (0.28) |
| 16S richness** | 221.15 (67.07) | 271.73 (47.31) |
| 16S Shannon** | 4.07 (0.52) | 4.46 (0.18) |

2 network analysis. Network nodes were represented by similar order level microbial groups (56% shared), while nodes were often different at the ASV level (20% shared) between clusters (Table S3). For both networks, only a handful of archaeal ASVs (five and six) were considered as nodes in the network analysis (Table S3; Fig. S5), reflecting their lower abundance in the 16S dataset (<2% on average) compared to Bacteria (Table S1). Protist-Protist associations contributed most to the overall number of edges in each network, followed by Prokaryote-Prokaryote and Prokaryote-Protist associations (Table 2). The number of Prokaryote-Prokaryote and Prokaryote-Protist associations increased by >100% in Cluster 1 (Table 2). For both networks, nearly half of Protist-Protist and Prokaryote-Protist edges were positive (47–58%); however, Prokaryote-Prokaryote edges were 84% and 87% positive in Cluster 1 and 2 networks, respectively (Table 2).

Edge density was slightly higher (and average path length lower) in Cluster 1 for the overall network or when networks were analyzed for each domain level pairing (Table 2). The exception were Prokaryote-Protist edges, which exhibited higher edge density and lower average path length in Cluster 2 (Table 2). Mean degree and closeness centrality were significantly higher (Wilcoxon tests, $p$-values $<0.001$) in Cluster 1 *vs.* 2 networks across all 16S or 18S ASVs used in the analysis (Figs. 4A–4B; Table S3). This pattern was conserved among the most relatively abundant microbial groups at the class level (Figs. S6–S7). Mean

**Table 2 Number of associations in each microbial network.** Number of microbial edges, nodes, edge density, and average path length for both Cluster 1 and 2 networks overall, as well as for networks based on major domain level associations (Protist-Protist, Prokaryote-Prokaryote, and Prokaryote-Protist). Proportion of edges that were positive (%) are shown for each type of association.

| Relationship type | Network | Number of edges | Number of nodes | Edge density | Average path length |
|---|---|---|---|---|---|
| Total | Cluster 1 | 1086 (66%) | 300 | 0.02 | 3.3 |
| | Cluster 2 | 707 (59%) | 297 | 0.02 | 4.3 |
| Protist-Protist | Cluster 1 | 501 (58%) | 150 | 0.05 | 3 |
| | Cluster 2 | 433 (52%) | 150 | 0.04 | 3.2 |
| Prokaryote-Prokaryote | Cluster 1 | 340 (84%) | 150 | 0.03 | 3.6 |
| | Cluster 2 | 167 (87%) | 147 | 0.02 | 6.5 |
| Prokaryote-Protist | Cluster 1 | 245 (58%) | 228 | 0.01 | 8.4 |
| | Cluster 2 | 107 (47%) | 150 | 0.17 | 3 |

degree was not significantly different between network clusters for Mamiellophyceae and Syndiniales (Fig. S7).

Though represented by similar order level microbes, the most prominent types of edges (both positive and negative) varied between networks (Fig. 5; Table S4), often becoming more prevalent in Cluster 1. Common Prokaryote-Prokaryote edges (mostly positive) that were higher in Cluster 1 *vs.* 2 included Flavobacteriales-SAR11, Flavobacteriales-Rhodobacterales, and Flavobacteriales-Flavobacteriales (Fig. 5A). Several Protist-Protist edges were higher in Cluster 2, including Bacillariophyta associated with Unassigned Dinophyceae, Dino-Groups I and II, Gonyaulacales, and Tintinnida (Fig. 5B). In contrast, associations between Bacillariophyta and Mamiellales, Cryptomonadales, and Peridiniales were elevated in Cluster 1 (Fig. 5B). Flavobacteriales-Bacillariophyta was the most prevalent cross-domain relationship in both networks and increased in Cluster 1 (Fig. 5C). Several cross-domain edges that were common in Cluster 1, including Rhodobacterales-Gymnodiniales, Rhodobacterales-Bacillariophyta, and Flavobacteriales-Strombidiida were not observed in the Cluster 2 network (Fig. 5C).

# DISCUSSION

Marine microbes interact with each other in a multitude of ways, influencing food web dynamics and the flow of energy in the ocean (*Worden et al., 2015*). Co-occurrence network analysis of amplicon metabarcoding data has become a widely used and powerful approach to characterize associations between microorganisms (*Faust & Raes, 2012*; *Faust, 2021*). Though providing ecological insight, network associations do not necessarily translate to causal interactions (*Blanchet, Cazelles & Gravel, 2020*). For instance, a positive (or negative) association between microbes may reflect an overlapping (or separate) niche (*Deutschmann et al., 2021*). Relationships generated from amplicon data should be verified by experimental testing and reinforced by searching for such interactions in primary literature and interaction databases (*Bjorbækmo et al., 2020*). Despite these drawbacks, network findings can be used to form (or build) hypotheses (*Santoferrara et al., 2020*) that

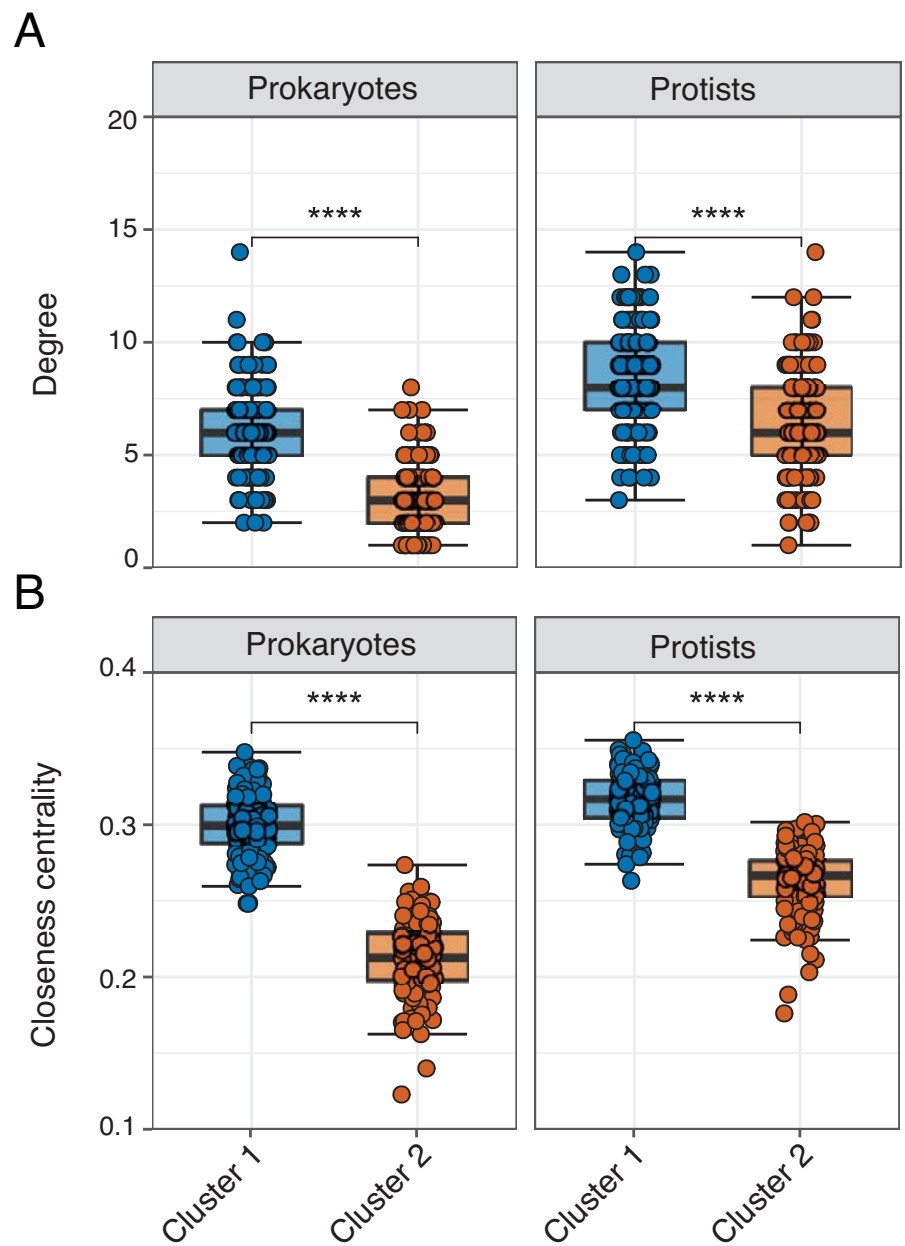

**Figure 4** **Network properties between clusters.** Faceted box plots displaying mean network degree (A) and closeness centrality (B) for prokaryotes and protists included in Cluster 1 (blue) and 2 (orange) networks. Degree and centrality were significantly different (Wilcoxon tests, **** = $p$-values < 0.0001) between clusters for prokaryotes and protists. Microbial ASVs (or nodes) within each network are shown as individual points ($n = 150$ per box plot). See Table S3 for raw data.

can be further tested in culture or in the field, advancing our knowledge of microbial interactions.

In our study, samples for network analysis were partitioned based on hierarchical clustering of 18S and 16S amplicon data, resulting in two environmentally distinct sample

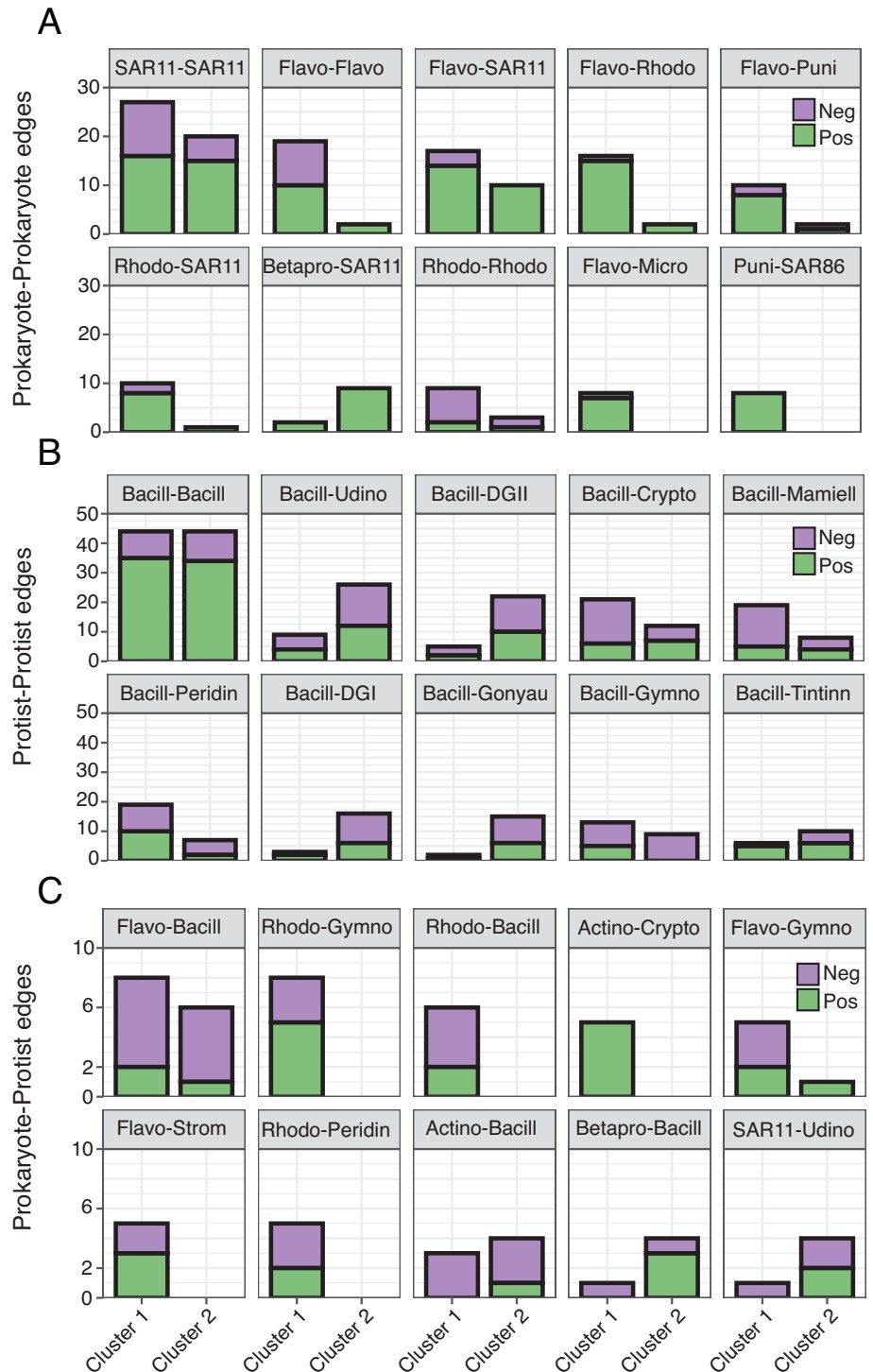

**Figure 5 Differences in the number of order level microbial associations between clusters.** (A) Prokaryote-Prokaryote, (B) Protist-Protist, (C) and Prokaryote-Protist relationships (faceted at order level) that were most prevalent (top ten) across both networks. (continued on next page...)

**Figure 5 (…continued)**
Number of positive (green) and negative (purple) network edges for each relationship are compared between clusters. Relationships that are absent from a given network were not detected. 16S abbreviations: Actino = Actinomarinales; Betapro = Betaproteobacteriales; Flavo = Flavobacteriales; Micro = Micrococcales; Puni = Puniceispiralles; Rhodo = Rhodobacterales; SAR11 = SAR11 Clade; SAR86 = SAR86 Clade. 18S abbreviations: Bacill = Bacillariophyta; Crypto = Cryptomonadales; DGI = Dino-Group I; DGII = Dino-Group II; Mamiell = Mamiellales; Gonyau = Gonyaulacales; Gymno = Gymnodiniales; Peridin = Peridiniales; Strom = Strombidiida; Tintinn = Tintinnida; Udino = Unassigned Dinophyceae. See Table S4 for a full list of network correlations.

sets: Cluster 1 = March-May/November-February *vs.* Cluster 2 = June-October. Several network properties, like network centrality, degree, and edge number, were higher in Cluster 1, despite networks having the same initial number of 16S (150) and 18S (150) ASVs (300 total) and being mainly (56%) represented by similar order level groups. Sample clusters reflected different environmental conditions, namely temperature, sunlight, $NO_3$, and $SiO_4$ (all lower in Cluster 1) that may have influenced network structure and microbial relationships. Temperature, sunlight, and nutrients are universal determinants of microbial metabolism, growth (or grazing), and community structure (*Hutchins & Fu, 2017*; *Logares et al., 2020*). Microbial physiology is often thought to scale with temperature in particular, though responses are species-specific and depend on other available resources, like nutrients (*Sarmento et al., 2010*; *Barton & Yvon-Durocher, 2019*). Applying these principles may also be challenging for microbial relationships, which are highly dynamic and vary over time and space in the ocean (*Chaffron et al., 2021*). Nevertheless, it remains important to better understand marine microbial networks and relationships, especially when considering the response of microbes to changing ecosystems.

## Microbial relationships vary between environmentally distinct periods

Prokaryote-Prokaryote and Prokaryote-Protist edges increased in number by >100% in Cluster 1, indicating these types of relationships may have been preferable in the estuary when conditions were less favorable for microbial growth and production (lower temperature, sunlight, and nutrients). Interestingly, ~85% of Prokaryote-Prokaryote edges were positive, which may represent mutualism among bacteria. This would be consistent with the stress gradient hypothesis, where facilitative associations among organisms increase under stressful conditions (*Bertness & Callaway, 1994*; *He & Bertness, 2014*). Prokaryote-Prokaryote associations often involved SAR11 and either Rhodobacterales or Flavobacteriales, the latter two being important for processing phytoplankton-derived organic carbon (*Buchan et al., 2014*). Oligotrophic (and genetically streamlined) microorganisms like SAR11 may rely on copiotrophs to assimilate sources of carbon or other metabolites needed for growth (*Buchan et al., 2014*; *Giovannoni, Thrash & Temperton, 2014*). In addition to lower temperature or sunlight, bacteria may have been impacted by depleted carbon sources. Diatoms were least relatively abundant in March-May (Cluster 1), and while winter-spring blooms can occur, phytoplankton biomass is typically lower at this time in the estuary (*Verity & Borkman, 2010*; *Anderson & Harvey, 2019*). Therefore, increased mutualism among heterotrophic bacteria may represent strategies to exchange

carbon, vitamins, or other metabolites and maximize growth when resources are otherwise limited and thermal conditions are less favorable.

The number of Prokaryote-Protist edges (both positive and negative) also increased in Cluster 1, especially heterotrophic bacteria (Flavobacteriales and Rhodobacterales) associated with diatoms, dinoflagellates, and ciliates. Diatom-bacteria associations are well known in the marine environment, spanning antagonistic, competitive, mutualistic, and symbiotic relationships (*Amin, Parker & Armbrust, 2012*; *Moran et al., 2012*; *Amin et al., 2015*; *Seymour et al., 2017*). Diatoms in our study were largely represented by chain-forming genera, like *Chaetoceros* and *Thalassiosira,* which form episodic blooms in the Skidaway River Estuary (*Anderson & Harvey, 2019*). Flavobacteriales and Rhodobacterales are known to closely associate with diatoms and rapidly exploit diatom-produced organic matter (*Buchan et al., 2014*). As observed among bacteria, positive diatom-bacteria relationships may reflect mutualism, with microbes exchanging metabolites and other compounds (*e.g.*, nutrients, vitamins, and hormones) required for growth (*Amin et al., 2015*). Negative diatom-bacteria associations may indicate resource competition or algicidal effects (*Amin, Parker & Armbrust, 2012*; *Meyer et al., 2017*), as well as other activities (*e.g.*, bacterial growth following consumption of plankton organic matter) that may have promoted niche separation. These potential relationships may have been more common among these groups during less favorable conditions in Cluster 1.

Other cross-domain edges were only present in Cluster 1, including Flavobacteriales and Rhodobacterales associated with heterotrophic (or mixotrophic) dinoflagellates, like Gymnodiniales (mainly *Gymnodinium*) and Peridiniales (mainly *Heterocapsa*), as well as the ciliate group Strombidiida (mainly *Strombidinium*). Negative associations between these taxa may indicate predation. Protist grazers are capable of ingesting bacteria (*Jeong et al., 2008*), utilizing alternative food sources when temperatures are lower (*Aberle et al., 2012*) or when their preferred algal prey is less abundant (*Paffenhöfer, Sherr & Sherr, 2007*). Measurable bacteria ingestion rates have been recorded for *Heterocapsa* and *Strombidinium* (2–34 cells protist$^{-1}$ hr$^{-1}$), though in general, ingestion by large protists is low compared to smaller flagellates (*Ichinotsuka, Ueno & Nakano, 2006*; *Kyeong et al., 2006*). Positive dinoflagellate-bacteria associations may represent symbiotic relationships, including photosymbiosis, nutrient fixation, or vitamin exchange (*Decelle, Colin & Foster, 2015*; *Bjorbækmo et al., 2020*). Yet, Prokaryote-Protist relationships remain largely unresolved in the ocean, especially at the species level (*Bjorbækmo et al., 2020*), which warrants further network studies inclusive of multiple domains of life and validation of such interactions in controlled lab (or field) experiments.

## Shifts in group-specific abundance between networks

Microbial networks may also vary over time and space due to changes in taxonomy and functional groups (*Stoecker & Lavrentyev, 2018*; *Vincent & Bowler, 2020*; *Chaffron et al., 2021*). While our networks were represented by similar taxa, the abundance of certain taxonomic groups changed, likely influencing network structure and relationships. Several groups, like Cryptomonadales, Peridiniales, Flavobacteriales, and Rhodobacterales, were more abundant and accounted for more network edges in Cluster 1. While impacted by

temperature, these groups were also correlated with ammonium ($NH_4$), which may have influenced their population dynamics and associations. Ammonium is a key nitrogen source in estuaries (*Damashek & Francis, 2018*) and evidence suggests that smaller phytoplankton, including cryptophytes and dinoflagellates, may better utilize $NH_4$ compared to larger phytoplankton, like diatoms, that prefer $NO_3$ (*Glibert et al., 2016*). Heterotrophic bacteria can also assimilate a substantial portion of $NH_4$ (>20%) in coastal environments (*Kirchman, 1994*). In the Skidaway River Estuary, increased levels of $NH_4$ (and other nutrients) have been recorded over decadal scales, resulting from increased anthropogenic activities (*Verity, 2002*; *Verity, Alber & Bricker, 2006*). As such, the role of dissolved nutrients in microbial networks should be considered, particularly in coastal areas with high nutrient runoff and potential for eutrophication or habitat loss.

Differences in network properties between clusters may have also been influenced by functional changes among dominant microbes. Certain protists that became more prevalent in Cluster 1, namely dinoflagellates and cryptophytes, are known to exhibit mixotrophy (*Stoecker et al., 2017*). Groups that can exploit both autotrophic and heterotrophic lifestyles may interact with a wider diversity of organisms, facilitating higher network connectivity (*Chaffron et al., 2021*). In general, we observed higher edge density and lower path length between nodes in Cluster 1 networks, which suggests that microbes were more connected to each other at this time. However, opposite trends were observed for Prokaryote-Protist edges, likely because cross-domain edges were spread out over more nodes in the Cluster 1 network. Despite fewer overall connections in Cluster 2, certain edges increased, including associations between diatoms and Syndiniales (Dino-Groups I and II), which are ubiquitous protist parasites in marine and estuarine ecosystems (*Guillou et al., 2008*). Increased parasite abundance and prevalence in Cluster 2 networks may have been driven by increased temperature, as parasitic infection is thought to be thermally influenced (*Park, Yih & Coats, 2004*). Though putative relationships, like infection (positive) or deterrence (negative), have not been empirically verified for Syndiniales and diatoms, these groups are often correlated in co-occurrence networks (*Sassenhagen et al., 2020*; *Vincent & Bowler, 2020*). Diatoms were also the most relatively abundant 18S group in our study (~25% on average), which may have explained their large contribution to Protist-Protist associations, including with Syndiniales.

## Considerations for co-occurrence analysis

An important consideration with co-occurrence network analysis is accounting for the presence of indirect edges that can contribute to dense ("hairball") networks and may cloud interpretation (*Röttjers & Faust, 2018*; *Faust, 2021*). Computational methods like SPIEC-EASI (used here) aim to account for indirect dependencies during network construction between microbes and promote sparser networks (*Kurtz et al., 2015*). Other recently developed programs, like EnDED (environmentally driven edge detection), are designed to reduce environmentally-driven associations after network construction, filtering for indirect dependency due to environmental factors (*Deutschmann et al., 2021*). To further improve network sparsity, we included only the most abundant 16S and 18S ASVs, a common strategy among co-occurrence network studies (*Faust, 2021*). This approach,

however, may limit detection of microbial correlations that are common in the community but involve less abundant taxa. For instance, though Archaea were present in our study, they exhibited low relative abundance in the 16S dataset (<2% on average) and contributed to <3% of network connections. It will be important to employ approaches to better characterize rare microbial relationships, while limiting the detection of indirect edges that may bias network analysis.

To assess environmental effects on networks, environmental variables are typically included as nodes in a single merged network (*Needham & Fuhrman, 2016*). Networks can also be constructed categorically (*e.g.*, by season) to reflect temporally variable conditions (*Kellogg et al., 2019*). With our approach, we separated samples for network analysis *a priori* based on beta diversity clustering, resulting in two environmentally distinct communities. Similar clustering of prokaryotes and protists allowed for merging of ASV tables in our study, though merging may not be possible in cases where different marker regions are distinctly clustered from each other. Even with our approach, collapsing weekly samples into one or two networks will undoubtedly mask network variability, underestimating ephemeral interactions that occur at hourly to weekly scales in dynamic estuaries (*Cloern, Foster & Kleckner, 2014*) and making it difficult to estimate network properties over a range of environmental factors. One idea would be to compare networks at a reasonable scale (*e.g.*, between months) in coastal areas that experience seasonal gradients in environmental factors. However, this would require sustaining ∼daily sampling frequency (as in *Needham & Fuhrman, 2016*; *Martin-Platero et al., 2018*) to maintain high sample to ASV ratios and avoid spurious correlations (*Faust, 2021*). Higher sampling frequency and binning of samples into more coarse time points would support correlations between network properties (*e.g.*, centrality and degree) and environmental variables, a strategy that has been employed with spatial samples (*Chaffron et al., 2021*).

## CONCLUSIONS

We explored correlation networks between two environmentally distinct microbial communities in a subtropical estuary. Instead of binning amplicon data arbitrarily, we used hierarchical cluster analysis to separate samples. With this approach, we observed higher network centrality, degree, and edge number for microbes in Cluster 1, even though environmental conditions were less favorable at this time (lower temperature, sunlight, and nutrients). Prokaryote-Prokaryote and Prokaryote-Protist edges increased the most in Cluster 1, while Protist-Protist associations were more stable. Though correlation networks present inferred associations (and not ecological interactions), differences in network properties may reflect changes in the types of relationships (*e.g.*, mutualism or competition) or shifts in taxonomic (or functional) prevalence in response to separate environmental periods. These findings represent an important step towards predicting microbial networks under varying conditions. Applying these clustering methods to new and existing amplicon surveys will help to broaden the scope of the analysis presented here (*e.g.*, single site and year), allowing for a deeper understanding of marine microbial networks, their relation to environmental factors, and potential sensitivity to anthropogenic ocean change.

## ACKNOWLEDGEMENTS

The authors would like to thank Tina Walters (UGA/SkIO) for help with initial PCR processing of metabarcoding samples and Margot Chisholm for assisting with bioinformatics of 16S samples. We thank Luke Thompson (NOAA/NGI) for helpful review of the manuscript. The authors thank Vera Tai and Ina Maria Deutschmann for their thoughtful review which improved the manuscript.

### Funding

This work is funded by a Sloan Research Fellowship to Elizabeth Harvey, and a National Science Foundation Grant (OCE-1831625) that supported the collaboration of Sean R Anderson, Margot Chisholm, and Elizabeth Harvey during the summer of 2020. The funders had no role in study design, data collection and analysis, decision to publish, or preparation of the manuscript.

### Grant Disclosures

The following grant information was disclosed by the authors:
A Sloan Research Fellowship to Elizabeth Harvey.
A National Science Foundation Grant: OCE-1831625.

### Competing Interests

The authors declare there are no competing interests.

### Author Contributions

- Sean R. Anderson conceived and designed the experiments, performed the experiments, analyzed the data, prepared figures and/or tables, authored or reviewed drafts of the article, and approved the final draft.
- Elizabeth L. Harvey conceived and designed the experiments, authored or reviewed drafts of the article, and approved the final draft.

### DNA Deposition

The following information was supplied regarding the deposition of DNA sequences:
   The amplicon sequences are available at GenBank: PRJNA575563 (18S rRNA) and PRJNA680039 (16S rRNA).

### Data Availability

   The R code and all files (ASV tables, metadata, network files) and raw metadata needed to run the code are available at GitHub https://github.com/sra34/SkIO-network.
   This project has been available at Zenodo: Anderson, Sean, & Harvey, Elizabeth. (2022). Microbial relationships in the Skidaway River Estuary (GA, USA) (Version 1) [Data set]. Zenodo. https://doi.org/10.5281/zenodo.6549350.

## Supplemental Information

Supplemental information for this article can be found online at http://dx.doi.org/10.7717/peerj.14005#supplemental-information.

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

# PeerJ

**Bowler C, De Vargas C, Eveillard D. 2021.** Environmental vulnerability of the global ocean epipelagic plankton community interactome. *Science Advances* **7**:1–16 DOI 10.1126/sciadv.abg1921.

**Chow CET, Kim DY, Sachdeva R, Caron DA, Fuhrman JA. 2014.** Top-down controls on bacterial community structure: microbial network analysis of bacteria, T4-like viruses and protists. *ISME Journal* **8**:816–829 DOI 10.1038/ismej.2013.199.

**Cloern JE, Foster SQ, Kleckner AE. 2014.** Phytoplankton primary production in the world's estuarine-coastal ecosystems. *Biogeosciences* **11**:2477–2501 DOI 10.5194/bg-11-2477-2014.

**Damashek J, Francis CA. 2018.** Microbial nitrogen cycling in estuaries: from genes to ecosystem processes. *Estuaries and Coasts* **41**:626–660 DOI 10.1007/s12237-017-0306-2.

**Decelle J, Colin S, Foster RA. 2015.** Photosymbiosis in marine planktonic protists. In: Ohtsuka S, Suzaki T, Horiguchi T, Suzuki N, Not F, eds. *Marine protists.* Tokyo: Springer Japan, 465–500 DOI 10.1007/978-4-431-55130-0_19.

**Deutschmann IM, Lima-Mendez G, Krabberød AK, Raes J, Vallina SM, Faust K, Logares R. 2021.** Disentangling environmental effects in microbial association networks. *Microbiome* **9**:1–18 DOI 10.1186/s40168-021-01141-7.

**Faust K. 2021.** Open challenges for microbial network construction and analysis. *ISME Journal* **15**:3111–3118 DOI 10.1038/s41396-021-01027-4.

**Faust K, Raes J. 2012.** Microbial interactions: from networks to models. *Nature Reviews Microbiology* **10**:538–550 DOI 10.1038/nrmicro2832.

**Fuhrman JA, Cram JA, Needham DM. 2015.** Marine microbial community dynamics and their ecological interpretation. *Nature Reviews Microbiology* **13**:133–146 DOI 10.1038/nrmicro3417.

**Fuhrman JA, Hewson I, Schwalbach MS, Steele JA, Brown MV, Naeem S. 2006.** Annually reoccurring bacterial communities are predictable from ocean conditions. *Proceedings of the National Academy of Sciences of the United States of America* **103**:13104–13109 DOI 10.1073/pnas.0602399103.

**Gilbert JA, Steele JA, Caporaso JG, Steinbrück L, Reeder J, Temperton B, Huse S, McHardy AC, Knight R, Joint I, Somerfield P, Fuhrman JA, Field D. 2012.** Defining seasonal marine microbial community dynamics. *ISME Journal* **6**:298–308 DOI 10.1038/ismej.2011.107.

**Giner CR, Balagué V, Krabberød AK, Ferrera I, Reñé A, Garcés E, Gasol JM, Logares R, Massana R. 2019.** Quantifying long-term recurrence in planktonic microbial eukaryotes. *Molecular Ecology* **28**:923–935 DOI 10.1111/mec.14929.

**Giovannoni SJ, Cameron Thrash J, Temperton B. 2014.** Implications of streamlining theory for microbial ecology. *ISME Journal* **8**:1553–1565 DOI 10.1038/ismej.2014.60.

**Glibert PM, Wilkerson FP, Dugdale RC, Raven JA, Dupont CL, Leavitt PR, Parker AE, Burkholder JM, Kana TM. 2016.** Pluses and minuses of ammonium and nitrate uptake and assimilation by phytoplankton and implications for productivity and community composition, with emphasis on nitrogen-enriched conditions. *Limnology and Oceanography* **61**:165–197 DOI 10.1002/lno.10203.

**Graff JR, Rynearson TA. 2011.** Extraction method influences the recovery of phyto-plankton pigments from natural assemblages. *Limnology and Oceanography: Methods* **9**:129–139 DOI 10.4319/lom.2011.9.129.

**Guillou L, Bachar D, Audic S, Bass D, Berney C, Bittner L, Boutte C, Burgaud G, De Vargas C, Decelle J, Del Campo J, Dolan JR, Dunthorn M, Edvardsen B, Holzmann M, Kooistra WHCF, Lara E, Le Bescot N, Logares R, Mahé F, Massana R, Montresor M, Morard R, Not F, Pawlowski J, Probert I, Sauvadet AL, Siano R, Stoeck T, Vaulot D, Zimmermann P, Christen R. 2013.** The protist ribosomal reference database (PR2): a catalog of unicellular eukaryote Small Sub-Unit rRNA sequences with curated taxonomy. *Nucleic Acids Research* **41**:D597–D604 DOI 10.1093/nar/gks1160.

**Guillou L, Viprey M, Chambouvet A, Welsh RM, Kirkham AR, Massana R, Scanlan DJ, Worden AZ. 2008.** Widespread occurrence and genetic diversity of marine parasitoids belonging to Syndiniales (Alveolata). *Environmental Microbiology* **10**:3349–3365 DOI 10.1111/j.1462-2920.2008.01731.x.

**He Q, Bertness MD. 2014.** Extreme stresses, niches, and positive species interactions along stress gradients. *Ecology* **95**:1437–1443 DOI 10.1890/13-2226.1.

**Hernandez DJ, David AS, Menges ES, Searcy CA, Afkhami ME. 2021.** Environmental stress destabilizes microbial networks. *The ISME Journal* **15**:1722–1734 DOI 10.1038/s41396-020-00882-x.

**Hu SK, Liu Z, Lie AAY, Countway PD, Kim DY, Jones AC, Gast RJ, Cary SC, Sherr EB, Sherr BF, Caron DA. 2015.** Estimating protistan diversity using high-throughput sequencing. *Journal of Eukaryotic Microbiology* **62**:688–693 DOI 10.1111/jeu.12217.

**Hutchins DA, Fu F. 2017.** Microorganisms and ocean global change. *Nature Microbiology* **2**:17058 DOI 10.1038/nmicrobiol.2017.58.

**Ibarbalz FM, Henry N, Brandão MC, Martini S, Busseni G, Byrne H, Coelho LP, Endo H, Gasol JM, Gregory AC, Mahé F, Rigonato J, Royo-Llonch M, Salazar G, Sanz-Sáez I, Scalco E, Soviadan D, Zayed AA, Zingone A, Labadie K, Ferland J, Marec C, Kandels S, Picheral M, Dimier C, Poulain J, Pisarev S, Carmichael M, Pesant S, Acinas SG, Babin M, Bork P, Boss E, Bowler C, Cochrane G, De Vargas C, Follows M, Gorsky G, Grimsley N, Guidi L, Hingamp P, Iudicone D, Jaillon O, Karp-Boss L, Karsenti E, Not F, Ogata H, Poulton N, Raes J, Sardet C, Speich S, Stemmann L, Sullivan MB, Sunagawa S, Wincker P, Pelletier E, Bopp L, Lombard F, Zinger L. 2019.** Global trends in marine plankton diversity across kingdoms of life. *Cell* **179**:1084–1097 e21 DOI 10.1016/j.cell.2019.10.008.

**Ichinotsuka D, Ueno H, Nakano SI. 2006.** Relative importance of nanoflagellates and ciliates as consumers of bacteria in a coastal sea area dominated by oligotrichous Strombidium and Strobilidium. *Aquatic Microbial Ecology* **42**:139–147 DOI 10.3354/ame042139.

**Jeong HJ, Seong KA, Yoo Y Du, Kim TH, Kang NS, Kim S, Park JY, Kim JS, Kim GH, Song JY. 2008.** Feeding and grazing impact by small marine heterotrophic dinoflag-ellates on heterotrophic bacteria. *Journal of Eukaryotic Microbiology* **55**:271–288 DOI 10.1111/j.1550-7408.2008.00336.x.

**Kandlikar GS, Gold ZJ, Cowen MC, Meyer RS, Freise AC, Kraft NJB, Moberg-Parker J, Sprague J, Kushner DJ, Curd EE. 2018.** Ranacapa: an R package and Shiny web app to explore environmental DNA data with exploratory statistics and interactive visualizations. *F1000Research* **7**:1734 DOI 10.12688/f1000research.16680.1.

**Kellogg CTE, McClelland JW, Dunton KH, Crump BC. 2019.** Strong seasonality in arctic estuarine microbial food webs. *Frontiers in Microbiology* **10**:2628 DOI 10.3389/fmicb.2019.02628.

**Kirchman DL. 1994.** The uptake of inorganic nutrients by heterotrophic bacteria. *Microbial Ecology* **28**:255–271 DOI 10.1007/BF00166816.

**Krabberød AK, Bjorbækmo MFM, Shalchian-Tabrizi K, Logares R. 2017.** Exploring the oceanic microeukaryotic interactome with metaomics approaches. *Aquatic Microbial Ecology* **79**:1–12 DOI 10.3354/ame01811.

**Kurtz ZD, Müller CL, Miraldi ER, Littman DR, Blaser MJ, Bonneau RA. 2015.** Sparse and compositionally robust inference of microbial ecological networks. *PLOS Computational Biology* **11**:1–25 DOI 10.1371/journal.pcbi.1004226.

**Kyeong AS, Hae JJ, Kim S, Gwang HK, Jung HK. 2006.** Bacterivory by co-occurring red-tide algae, heterotrophic nanoflagellates, and ciliates. *Marine Ecology Progress Series* **322**:85–97 DOI 10.3354/meps322085.

**Lambert S, Lozano JC, Bouget FY, Galand PE. 2021.** Seasonal marine microorganisms change neighbours under contrasting environmental conditions. *Environmental Microbiology* **23**:2592–2604 DOI 10.1111/1462-2920.15482.

**Lima-Mendez G, Faust K, Henry N, Decelle J, Colin S, Carcillo F, Chaffron S, Ignacio-espinosa JC, Roux S, Vincent F, Bittner L. 2015.** Determinants of community structure in the grobal plankton interactome. *Science* **348**:1262073 DOI 10.1126/science.1262073.

**Liu H, Roeder K, Wasserman L. 2010.** Stability approach to regularization selection (StARS) for high dimensional graphical models. *Advanced Nerual Information Process System* **24**:1432–1440.

**Logares R, Deutschmann IM, Junger PC, Giner CR, Krabberød AK, Schmidt TSB, Rubinat-Ripoll L, Mestre M, Salazar G, Ruiz-González C, Sebastián M, De Vargas C, Acinas SG, Duarte CM, Gasol JM, Massana R. 2020.** Disentangling the mechanisms shaping the surface ocean microbiota. *Microbiome* **8**:1–17 DOI 10.1186/s40168-020-00827-8.

**Martin-Platero AM, Cleary B, Kauffman K, Preheim SP, McGillicuddy DJ, Alm EJ, Polz MF. 2018.** High resolution time series reveals cohesive but short-lived communities in coastal plankton. *Nature Communications* **9**:1–11 DOI 10.1038/s41467-017-02571-4.

**McMurdie PJ, Holmes S. 2013.** phyloseq: an R package for reproducible interactive analysis and graphics of microbiome census data. *PLOS ONE* **8**:e61217 DOI 10.1371/journal.pone.0061217.

**Meyer N, Bigalke A, Kaulfuß A, Pohnert G. 2017.** Strategies and ecological roles of algicidal bacteria. *FEMS Microbiology Reviews* **41**:880–899 DOI 10.1093/femsre/fux029.

**Milici M, Deng ZL, Tomasch J, Decelle J, Wos-Oxley ML, Wang H, Jáuregui R, Plumeier I, Giebel HA, Badewien TH, Wurst M, Pieper DH, Simon M, Wagner-Döbler I. 2016.** Co-occurrence analysis of microbial taxa in the Atlantic ocean reveals high connectivity in the free-living bacterioplankton. *Frontiers in Microbiology* **7**:1–20 DOI 10.3389/fmicb.2016.00649.

**Moran MA, Reisch CR, Kiene RP, Whitman WB. 2012.** Genomic insights into bacterial DMSP transformations. *Annual Review of Marine Science* **4**:523–542 DOI 10.1146/annurev-marine-120710-100827.

**Needham DM, Fuhrman JA. 2016.** Pronounced daily succession of phytoplankton, archaea and bacteria following a spring bloom. *Nature Microbiology* **1**:16005 DOI 10.1038/nmicrobiol.2016.5.

**Needham DM, Sachdeva R, Fuhrman JA. 2017.** Ecological dynamics and co-occurrence among marine phytoplankton, bacteria and myoviruses shows microdiversity matters. *ISME Journal* **11**:1614–1629 DOI 10.1038/ismej.2017.29.

**Oksanen J, Blanchet FG, Friendly M, Kindt R, Legendre P, McGlinn D, Minchin PR, O'Hara RB, Simpson GL, Solymos P, Stevens MHH, Szoecs E, Wagner H. 2018.** vegan: community ecology package. R package version 2.5-2. *Available at https://cran.r-project.org/web/packages/vegan/index.html*.

**Paffenhöfer GA, Sherr BF, Sherr EB. 2007.** From small scales to the big picture: persistence mechanisms of planktonic grazers in the oligotrophic ocean. *Marine Ecology* **28**:243–253 DOI 10.1111/j.1439-0485.2007.00162.x.

**Parada AE, Needham DM, Fuhrman JA. 2016.** Every base matters: assessing small subunit rRNA primers for marine microbiomes with mock communities, time series and global field samples. *Environmental Microbiology* **18**:1403–1414 DOI 10.1111/1462-2920.13023.

**Park MG, Yih W, Coats DW. 2004.** Parasites and phytoplankton, with special emphasis on dinoflagellate infections. *Journal of Eukaryotic Microbiology* **51**:145–155 DOI 10.1111/j.1550-7408.2004.tb00539.x.

**Piccardi P, Vessman B, Mitri S. 2019.** Toxicity drives facilitation between four bacterial species. *Proceedings of the National Academy of Sciences of the United States of America* **116**:15979–15984 DOI 10.1073/pnas.1906172116.

**Pruesse E, Quast C, Knittel K, Fuchs B, Ludwig W, Peplies J, Glöckner FO. 2007.** SILVA: a comprehensive online resource for quality checked and aligned ribosomal RNA sequence data compatible with ARB. *Nucleic Acids Research* **35**:7188–7196 DOI 10.1093/nar/gkm864.

**R Core Team. 2020.** A language and environment for statistical computing. Version 3.6.3. Vienna: R Foundation for Statistical Computing. *Available at https://www.r-project.org*.

**Röttjers L, Faust K. 2018.** From hairballs to hypotheses–biological insights from microbial networks. *FEMS Microbiology Reviews* **42**:761–780 DOI 10.1093/femsre/fuy030.

**Rousseeuw PJ. 1987.** Silhouettes: a graphical aid to the interpretation and validation of cluster analysis. *Journal of Computational and Applied Mathematics* **20**:53–65 DOI 10.1016/0377-0427(87)90125-7.

Santoferrara L, Burki F, Filker S, Logares R, Dunthorn M, McManus GB. 2020. Perspectives from ten years of protist studies by high-throughput metabarcoding. *Journal of Eukaryotic Microbiology* **67**:612–622 DOI 10.1111/jeu.12813.

Sarmento H, Montoya JM, Vázquez-Domínguez E, Vaqué D, Gasol JM. 2010. Warming effects on marine microbial food web processes: How far can we go when it comes to predictions? *Philosophical Transactions of the Royal Society B: Biological Sciences* **365**:2137–2149 DOI 10.1098/rstb.2010.0045.

Sassenhagen I, Irion S, Jardillier L, Moreira D, Christaki U. 2020. Protist interactions and community structure during early autumn in the kerguelen region (Southern Ocean). *Protist* **171**:1–20 DOI 10.1016/j.protis.2019.125709.

Seymour JR, Amin SA, Raina JB, Stocker R. 2017. Zooming in on the phycosphere: the ecological interface for phytoplankton-bacteria relationships. *Nature Microbiology* **2**:17065 DOI 10.1038/nmicrobiol.2017.65.

Shannon P, Markiel A, Ozier O, Baliga N, Wang J, Ramage D, Amin N, Schwikowski B, Ideker T. 2003. Cytoscape: a software environment for integrated models of biomolecular interaction networks. *Genome Research* **13**:2498–2504 DOI 10.1101/gr.1239303.metabolite.

Sogin ML, Morrison HG, Huber JA, Welch DM, Huse SM, Neal PR, Arrieta JM, Herndl GJ. 2006. Microbial diversity in the deep sea and the underexplored rare biosphere. *Proceedings of the National Academy of Sciences of the United States of America* **103**:12115–12120 DOI 10.1073/pnas.0605127103.

Stoeck T, Bass D, Nebel M, Christen R, Jones MD, Breiner HW, Richards TA. 2010. Multiple marker parallel tag environmental DNA sequencing reveals a highly complex eukaryotic community in marine anoxic water. *Molecular Ecology* **19**:21–31 DOI 10.1111/j.1365-294X.2009.04480.x.

Stoecker DK, Hansen PJ, Caron DA, Mitra A. 2017. Mixotrophy in the marine plankton. *Annual Review of Marine Science* **9**:311–335 DOI 10.1146/annurev-marine-010816-060617.

Stoecker DK, Lavrentyev PJ. 2018. Mixotrophic plankton in the polar seas: a pan-Arctic review. *Frontiers in Marine Science* **5**:292 DOI 10.3389/fmars.2018.00292.

Sunagawa S, Coelho LP, Chaffron S, Kultima JR, Labadie K, Salazar G, Djahanschiri B, Zeller G, Mende DR, Alberti A, Cornejo-Castillo FM, Costea PI, Cruaud C, D'Ovidio F, Engelen S, Ferrera I, Gasol JM, Guidi L, Hildebrand F, Kokoszka F, Lepoivre C, Lima-Mendez G, Poulain J, Poulos BT, Royo-Llonch M, Sarmento H, Vieira-Silva S, Dimier C, Picheral M, Searson S, Kandels-Lewis S, Bowler C, De Vargas C, Gorsky G, Grimsley N, Hingamp P, Iudicone D, Jaillon O, Not F, Ogata H, Pesant S, Speich S, Stemmann L, Sullivan MB, Weissenbach J, Wincker P, Karsenti E, Raes J, Acinas SG, Bork P, Boss E, Bowler C, Follows M, Karp-Boss L, Krzic U, Reynaud EG, Sardet C, Sieracki M, Velayoudon D. 2015. Structure and function of the global ocean microbiome - SM. *Science* **348**:1261359–1261359 DOI 10.1126/science.1261359.

**Tipton L, Müller CL, Kurtz ZD, Huang L, Kleerup E, Morris A, Bonneau R, Ghedin E. 2018.** Fungi stabilize connectivity in the lung and skin microbial ecosystems. *Microbiome* **6**:1–14 DOI 10.1186/s40168-017-0393-0.

**Verity PG. 2002.** A decade of change in the skidaway river estuary. I. Hydrography and nutrients. *Estuaries* **25**:944–960 DOI 10.1007/BF02691343.

**Verity PG, Alber M, Bricker SB. 2006.** Development of hypoxia in well-mixed sub-tropical estuaries in the Southeastern USA. *Estuaries and Coasts* **29**:665–673 DOI 10.1007/BF02784291.

**Verity PG, Borkman DG. 2010.** A decade of change in the skidaway river estuary. III. Plankton. *Estuaries and Coasts* **33**:513–540 DOI 10.1007/s12237-009-9208-2.

**Vincent F, Bowler C. 2020.** Diatoms are selective segregators in global ocean planktonic communities. *MSystems* **5**:1–14 DOI 10.1128/mSystems.00444-19.

**Ward CS, Yung CM, Davis KM, Blinebry SK, Williams TC, Johnson ZI, Hunt DE. 2017.** Annual community patterns are driven by seasonal switching between closely related marine bacteria. *ISME Journal* **11**:1412–1422 DOI 10.1038/ismej.2017.4.

**Weiss S, Xu ZZ, Peddada S, Amir A, Bittinger K, Gonzalez A, Lozupone C, Zaneveld JR, Vázquez-Baeza Y, Birmingham A, Hyde ER, Knight R. 2017.** Normalization and microbial differential abundance strategies depend upon data characteristics. *Microbiome* **5**:27 DOI 10.1186/s40168-017-0237-y.

**Wickham H, Averick M, Bryan J, Chang W, McGowan LD, François R, Grolemund G, Hayes A, Henry L, Hester J, Kuhn M, Pedersen TL, Miller E, Bache SM, Müller K, Ooms J, Robinson D, Seidel DP, Spinu V, Takahashi K, Vaughan D, Wilke C, Woo K, Yutani H. 2019.** Welcome to the tidyverse. *Journal of Open Source Software* **4**:1686–1681 DOI 10.21105/joss.01686.

**Worden AZ, Follows MJ, Giovannoni SJ, Wilken S, Zimmerman AE, Keeling PJ. 2015.** Rethinking the marine carbon cycle: factoring in the multifarious lifestyles of microbes. *Science* **347**:127594 DOI 10.1126/science.1257594.

**Xia X, Guo W, Liu H. 2017.** Basin scale variation on the composition and diversity of archaea in the Pacific Ocean. *Frontiers in Microbiology* **8**:1–15 DOI 10.3389/fmicb.2017.02057.