# Peer review of "Estuarine microbial networks and relationships vary between environmentally distinct communities"

_PeerJ, doi:10.7717/peerj.14005_

## Round 0.1 · original submission · Major Revisions

I agree with the reviewers that some major revisions are required before this manuscript can be considered further for publication. Please carefully address each of the reviewers' comments when resubmitting your manuscript. Of primary concern are how sampling and subsampling were conducted, how ASVs were filtered, and what/why are the assumptions made when assessing networks all need to be thoroughly addressed.

Reviewer 1 ·

Basic reporting

The authors used clear language throughout the manuscript and include recent and relevant literature references. Raw data is shared and deposited. Moreover, processed data and scripts are shared on a Github repository, which is great. It would be good if that repository could also be submitted to a more permanent repository, like Zenodo, so continued access to the source code listed in the article can be guaranteed!
The figures presented in the manuscript are of high quality and with well-written legends.
Some of the hypotheses presented in the discussion are not fully supported by the analyses presented by the authors.

Experimental design

The authors collected samples on 33 sampling days and used this to infer networks with SpiecEasi. Those networks were then analyzed in Cytoscape. However, the supplementary data shows there are 96 samples in total. The authors need to clarify how replicates were collected. If these were technical replicates, and they are included in network inference, this is pseudoreplication. The authors also make no mention of blanks or contaminants throughout the manuscript.

Validity of the findings

The processing of the sequencing data is appropriate and all data has been provided. If there is no pseudoreplication, this study includes sufficient samples to support its results. Network inference was also performed with a state-of-the-art method. However, the findings presented in the discussion, suggesting that these networks may in some way be linked to microbial interactions, are not fully supported by the results.

Additional comments

The authors collected surface water samples for 12 months, at 33 different sampling days, at the Skidaway River Estuary. They acquired 16S and 18S rRNA sequences from these samples and used the relative abundances of ASVs to infer networks. To minimize the effect of drivers of network structure, they inferred one network per cluster (2 clusters in total). They then compare network properties of these networks.

Main comments

The current wording of the manuscript suggests that networks were inferred using all ASVs, not just the top 150 bacterial and top 300 protist ASVs. Did the authors filter the ASV table before network inference, and if so, did they preserve the total relative abundance?

Why did the authors choose to focus on the top ASVs only? This decision is not justified anywhere in the manuscript. The authors should clarify how large the networks were before this filtering step. Moreover, associations may not necessarily take place between ASVs, but rather between functional guilds (e.g. groups of ASVs competing for the same resources). It is possible for a group of ASVs to vary wildly in abundance individually, but have stable abundances as a group (Louca et al. 2017).

The authors claim they worked with clustered data to reduce the impact of environmental variables and apply no other methods to remove environmentally-drive associations from their networks. However, there may still be very large differences between sampling points, considering samples were weeks apart. The authors cite EnDED in their discussion; applying a method like this would make their conclusions more robust (Deutschmann et al. 2021).

Figure 2 suggests that differences in beta-diversity were larger for cluster 1 networks than they were for cluster 2 networks. Consequently, it seems likely that cluster 1 samples may simply contain a lot more ASVs and have a larger effective sampling size (more unique communities) compared to cluster 2 samples, leading to a larger network. Moreover, network size is correlated to several network properties. The authors should try to control for this effect by controlling the ASV number and sample number used for network inference.

The authors discuss potential microbial interactions extensively in the discussion. However, inferred associations cannot be linked to interactions so easily (Blanchet et al. 2020). For example, the authors suggest that there is a high degree of mutualism in bacteria. However, bacteria may also co-occur and therefore have positive associations if they are competing in the same niche. Similarly, the authors suggest that there is increased competition between diatoms and bacteria. This may also be due to niche filtering, as environmental drivers of community structure was not taken into account. The presentation of associations as potential interactions, with some minor caveats, is misleading, since the authors did not sample at a timescale where we would expect to see these interactions take place.

Minor comments

51: Despite less favorable conditions and lower diversity, network properties (centrality, degree, and edge number) increased in Cluster 1.
* * *
This suggests that network properties are linked to favorable conditions, but this is not the case. The first part of the sentence should be removed.

56: We provide evidence of increased mutualistic, competitive, and predatory relationships in Cluster 1, possibly indicative of resource acquisition under less favorable conditions.
* * *
These networks are not evidence of these interactions.

106: The most common strategy to assess environmental impact has been to aggregate samples into a single network, including environmental variables as additional nodes that can be correlated to ASVs (Fuhrman, Cram & Needham, 2015; Needham, Sachdeva & Fuhrman, 2017). However, this approach often results in fewer environmental network correlations compared to those between species (Gilbert et al., 2012; Chow et al., 2014).
* * *
The authors suggest that the loss of correlations is negative. That is not the case if those correlations are false positives, as suggested by the supplementary analysis done by Tackmann (2019).

358: Microbial interactions in response to less favorable conditions
* * *
The heading should be changed since no microbial interactions were measured. This entire section should be rewritten, since it is based on the assumption that co-occurrence relationships reflect interactions, which is not necessarily true (Blanchet et al. 2020).

References

Blanchet, F. G., Cazelles, K., & Gravel, D. (2020). Co‐occurrence is not evidence of ecological interactions. Ecology Letters, 23(7), 1050-1063.
Deutschmann, I. M., Lima-Mendez, G., Krabberød, A. K., Raes, J., Vallina, S. M., Faust, K., & Logares, R. (2021). Disentangling environmental effects in microbial association networks. Microbiome, 9(1), 1-18.
Louca, S., Jacques, S. M., Pires, A. P., Leal, J. S., Srivastava, D. S., Parfrey, L. W., ... & Doebeli, M. (2016). High taxonomic variability despite stable functional structure across microbial communities. Nature ecology & evolution, 1(1), 1-12.
Tackmann, J., Rodrigues, J. F. M., & von Mering, C. (2019). Rapid inference of direct interactions in large-scale ecological networks from heterogeneous microbial sequencing data. Cell systems, 9(3), 286-296.

·

Basic reporting

no comment

Experimental design

The research question and the data collected is relevant and meaningful.

Additional details on methodology, particularly in describing the primers used in the PCRs and which taxa were included or excluded from the analyses.

Detailed comments on the methods:

Lines 166 to 170 - Clarify the primers used for the first PCR - likely has adapter sequences on the primer in addition to the target-specific regions.

Line 177 - Similarly, the second PCR has adapter regions to add in addition to sample-specific indices

More clearly describe how the data was processed, which taxa were kept or excluded to result in the data that was analyzed further (see comments for Line 188 and 201 below.

Line 188 - 16S V4 ASVs could include Bacteria and Archaea. At this point in the analysis, what is the frequency of Archaea? Have these been excluded from the analysis so that the focus is on Bacteria?

Line 199 - be specific about which packages were used (not "using packages like...")

Line 201 - Be specific about which Eukaryotic groups were excluded, and more precisely define what are considered protists in this analysis. Was it only Metazoa and Streptophyta, or were there others?

Line 202 - as discussed in the next comment for Lines 207-208, the name of the taxonomic categories need clarification. For Bacteria, ASVs that could not be identified as Bacteria would be the Domain level, not Kingdom.

Lines 207-208 - This line implies that Table S1 contains information on all ASVs, but the table contains ~ 10,000 ASVs from eukaryotes, and none from Bacteria and Archaea. Please clarify, and include the Bacteria and Archaea data if that was the intent. For the taxonomic categories in Table S1, the taxa labeled as Kingdom (i.e. Eukaryota) should be labeled as "Domain". Division should be "Kingdom", Class should be "Phylum". However, for taxa labeled as Order and Family - the category names for these taxa are not consistent across the eukaryotes (e.g. in the same column - Peridiniales is considered an Order where as Crustacea is often considered a Class (or Subphylum)). Probably better to label the categories as Ranks based on the PR2 database, i.e. Rank 1, Rank 2, Rank 3, etc...


Line 210 - Often, singletons (or even more frequent ASVs) are removed after sequence filtering and identification of ASVs to account for potential sequencing errors. Was this performed during data processing? What is the reason for removing ASVs here, after rarefying the data?

Validity of the findings

The interpretation of the network analyses was not always justified. This may be due to a lack of details in presenting the results, but a more thorough consideration of the type of data analyzed (e.g. the top 300 taxa) and how this set of taxa influences the analysis and the conclusions is needed. Other aspects of the analysis need to be more clearly explained (sometimes the description of the results are not correct), but these are relatively less major in nature.

Major comments on the validity of the findings:

Lines 296-297 and Table 1 - The description of the dbRDA results are confusing, perhaps because they are not being interpreted quite correctly. For example, for temperature, the dbRDA scores indicate the coordinates of the vector tip on the biplot. The interpretation of the vector depends on the position of the samples in the biplot. In this case, since Cluster 1 and Cluster 2 samples generally separate in the same direction as the temperature vector and since the temperature vector is relatively long and points in the direction of Cluster 2 samples, this indicates that higher temperatures is a relatively important factor in distinguishing Cluster 2 samples from Cluster 1. In contrast, lower NH4 (rather than explained as a negative correlation) is an explanatory factor for the change in composition from Cluster 1 to Cluster 2 samples. The ANOVA tests the significance of the environmental contraints on the dbRDA, not the significance of the scores.

Table 1 - Indicate why dbRDA scores for the first axis only are reported, presumably because Cluster 1 and Cluster 2 samples are separated by the first axis in the biplot. But this table is not necessary since environmental factor vectors are plotted in the dbRDA biplot. In interpreting these vectors, dbRDA scores are not a measure of correlation, but indicate the coordinates for tip of the vector (arrow) where a longer vector indicates the importance of this factor relative to the other factors. The sign of the dbRDA scores reflects the direction in which the environmental factor separates the samples, i.e. vector points towards increasing values of the environmental factor.

Line 243 and Line 315 - What diversity of taxa were included in the network? Is there a difference in the composition of the most relative abundant taxa in Cluster 1 compared to Cluster 2? Was there a difference in the number of bacteria and protist taxa out of the 300 assessed in the 2 networks? Wouldn't that proportion affect the relative number of edges detected between the bac-bac, bac-prot, and prot-prot? e.g. if there were more bacterial taxa, then the number of observed of bac-bac associations would be higher. What about examining taxa that have the highest degree, and the taxa that are joined to these hubs?

Lines 359-360, Lines 469-471 - What is the basis for attributing higher connectivity and edge number to bac-bac and bac-protist connections? This needs to be more clearly explained. Figure 5 is a selected subset of connections, and does not convincingly demonstrate these trends. In considering protist-protist edges, there are more protist-protist edges, and also higher number in Cluster 1 for some protist-protist taxa, so not clear how this is different from bac-bac and bac-protist edges. If this conclusion is just based on positive edges, the described trend also does not seem to be clearly evident. Please explain. And related to comment for Line 243, have you assessed whether the combination of taxa used will affect the number of bac-bac edges and bac-protist edges? E.g. Does the composition of the 300 taxa results in a biased count of these edges in comparing Cluster 1 to Cluster 2 networks, i.e. if there are more bacteria in the 300 taxa, does that mean that more bac-bac edges will be counted? At least a consideration of this issue should be discusses, as in the nicely written discussion on co-occurence analyses generally (Lines 436-464).

Lines 390-391 - If these relationships indicate predation, then these would be represented by negative edges. Please clarify or expand on this discussion as both positive and negative edges were observed for these associations. On the other hand most Bacillariophyta-Flavobacterales edges were negative - so can a predatory relationship be assumed here? As degraders of complex carbohydrates, Flavobacteria may increase as a diatom bloom in in decline? As discussed (Lines 396-399), relationships remain poorly defined, but some additional hypotheses could be addressed.

Lines 425 and 426 - The high number of edges to diatoms is likely because diatoms comprise a large fraction of the protist data (~ 30%). So when the top 300 taxa were chosen for the network analysis, many of the protists will be diatoms, and all of the interactions counted in Table 5 are with diatoms. So the statement "especially with diatoms" needs to be better considered. Need to consider or make note that interactions with other rarer taxa were not observed because they were not included in the analysis.

Additional comments

Additional comments:

Line 166 - here, and throughout - capitalize Bacteria and Archaea.

Line 258 - for class-level, specify the classification ranks that match "class-level" in this analysis. Is it 4th rank for PR2 classification of eukaryotes, but 3rd rank for Silva classification of Bacteria?
See comment for Table S2 as well - Referring to the taxonomic ranks as Class or Order is not always correct as these ranks in taxonomic databases do not consistently refer to classes, orders, or families depending on the group. Would be better to refer to these different taxonomic levels by their Rank number in the database (i.e. Rank 4 can refer to Order or Class level taxa). Or, put "class-level" in parentheses, and (i.e. 4th Rank).

Line 261 - similar comment as for line 258.

Table S2 - include units for each measurement. Order the rows in chronological order, i.e. 2018 samples after 2017.

Lines 266-267 - Would not say that PO4, salinity, PON, and POC are stable. These environmental factors also vary, but perhaps PO4 and salinity do not vary as much as was expected. But, the more important result to communicate is that these variations are not strongly correlated with temp. (and therefore seasonality)

Line 282 and elsewhere - Unassigned Dinophyceae is not a cohesive group reflecting ASVs with a shared evolutionary history. This is a catch all description for ASVs identified as belonging to Dinophyceae, but not assigned to a lower taxonomic group. Collectively these ASVs may not belong to the same taxa, and may be divergently related. So, although their relative abundance is correlated with temperature, it is difficult to interpret what this might mean ecologically as they are likely not an evolutionarily cohesive set of ASVs...unless their evolutionary relationships of these ASVs have been examined phylogenetically?

Line 287 - Here and elsewhere - the Bacillariophyta_X is an unnamed taxonomic category within the Bacillariophyta (diatoms), i.e. there is not a name assigned to this rank for the diatoms. In the PR2 database, all taxa in the Bacillariophyta_X are grouped in the higher category Bacillariophyta, so these can safely just be referred to as Bacillariophyta.

Line 291 - Since NH4 has the opposite trend with temperature, this is not surprising.

Lines 406 - There groups "were" more abundant in Cluster 1 (i.e. winter months), not "became more abundant".

Line 412 - correct spelling to "Gilbert"


Figure 1. The top ten most prevalent across both networks. Please clarify - does this consider a union of the top ten in each network (i.e. for each bar graph, counting top 10 co-occurrences in cluster 1 plus top 10 in cluster 2), or the top ten considering of all samples together.

Figure 1A - not clear what is meant by: "Abundance data is shown in triplicate". For some months, there are only two points, how is this in triplicate? What do the points represent?

Figure 1B - Provide a scale bar for the diameter of the circles.

Figure 2 - similar comments as Figure 1, to better describe the data in the figures

Figure 5 - Clarify the analysis presented in the figure. What is the criteria for "Prominent microbial relationships", what make these prominent? Perhaps better titled as "Number of significant microbial co-occurrences within a cluster" ?

Figure 5C - Consistently label graphs as Bacteria group first then protist.

Provide legends for supplemental figures and tables.

·

Basic reporting

Anderson et al. explore microbial systems at the Skidaway River Estuary by contrasting association networks inferred from two environmentally distinct communities. Microbial systems are dynamic, and their community composition is determined through a combination of ecological processes, including selection via environmental factors. Microbial interactions, and their response to environmental changes, are barely known. The present work contributes to elucidating the natural heterogeneity of microbial networks throughout the year. The study will become valuable for future long-term studies that need to distinguish long-term changes/shifts (e.g., due to climate change) vs. seasonal ones regularly occurring throughout the year. Further, the study underlines the need for sustained monitoring of microbial systems.

The manuscript uses clear language, and the paragraphs form a coherent story. I enjoyed reading the well-referenced introduction and discussion. However, the repeated misusage of the network terminology needs to be corrected (see comments to network terminology). The methods are well chosen. For example, the authors used a network construction tool that takes the compositional nature of microbial data into account and aims to infer sparse networks without indirect dependencies. Yet, some clarifications should be made (see major comments). Data and code have been shared publicly. All Figures and Tables, including supplementary ones, have been described in the online system. However, the descriptions of supplementary Figures and Tables have been missing in the downloaded version.

Experimental design

Major comments
- Line 137-138: [changes in the types of relationships] – Do relationships change because different top 300 ASVs were used for the two clusters? It was not clearly stated in the method section (Lines 243-244): Were the 300 selected ASVs the same for both clusters? If yes: Do they equally align with the most abundant ASVs in either cluster? If no: Are relationships observed in cluster 1 but not in cluster 2 due to no association being determined or because the ASVs were not included to infer an association.
- Line 149: 33 sampling days < number of samples. How many samples in total? At which step or steps were replicates obtained? Are the number of repeats comparable across sampling days? Also, when only monthly sampling was available, have weekly samples not been planned or missed due to technical reasons?
- Line 166: Archaea are mentioned here but not in downstream analysis, e.g., in Lines 186-189 (taxonomy inference), 199 (community dynamics), 222 (group-specific abundance), and 243-244 (network construction). Have they been removed?
- Line 201-204: “several groups” – please specify which groups were removed apart from Metazoa and Streptophyta. If you refer to the list following the sentence, stating ‘several groups’ seems misleading. ASVs assigned to chloroplasts or mitochondria were removed – did the data contain ASVs corresponding to plastids, which have been removed?
- Line 209: Were both tables rarefied to the same minimum?
- Line 209-213: No reasons nor references given on choice of steps.
- Lines 265-266: Has such analysis been done for all pairs of environmental factors? If not, why only for that one pair? In Line 291: “interestingly” – I do not find it interesting since it was stated in the 1st Result paragraph that temperature peaked June-Oct and NH4 in Dec-Feb.
- Line 404: similar [core] taxa – “core taxa” has not been defined in methods nor presented in results. The statement of networks being represented by similar taxa is supported by Figure 5. The Figure is valuable, indicating the microbial association differences between the two clusters for specific groups. However, the work would benefit by quantifying how many nodes and associations are the same between clusters. Have associations switched signs? How many microbes (present throughout the year) keep their association partners vs. switch association partners?

Validity of the findings

Reproducibility
- The authors made their data and code freely available: read sequences (NCBI), well-documented R script including commands for data visualization, ASV tables, metadata, and the Cytoscape network (GitHub).
- The files ‘18S-rooted-tree.qza” and “16S-rooted-tree.qza” are missing but used in the R script, e.g., l. 34-48, 58, 237, and 249.
- The given framework perfectly results in the same months used for network construction for bacteria and protists. However, this may not happen in studies that apply the presented approach. In my opinion, it is not needed to elaborate deeply on this, but it could be mentioned briefly.

Additional comments

Network terminology
- Lines 60-62: agreed. However, I suggest exchanging [networks] for [communities] or [quantities]. The microbial communities are monitored by estimating their abundances. Abundance tables are used to infer association networks. We cannot monitor the networks; we aim to infer microbial relationships via association studies.
- Lines 110-112, 133, 243-244, 345, 453: separating/filtering/partitioning [networks] – [networks] were not separated/filtered/partitioned but the [data tables]/[ASVs] used for network construction. Using the same number of ASVs in both groups for network construction is important for comparing the two networks, especially when inferring the network as a whole as SPIEC-EASI does. One approach could have been to construct one single network, and then microbes of specific seasons potentially could have ended up in the same cluster (“network clustering”). In contrast, your study constructs two group-specific microbial networks. It is not straightforward that the networks (their network properties) should change. That is why your study is an excellent contribution to characterizing microbial networks.
- Line 138-140: [network clustering] – this term should be avoided since it refers to a graph theoretic task. Such network clustering has not been done in this study nor referenced in Line 114. Examples of network clustering (also known as community or module detection) include the Girvan-Newman algorithm (Girvan & Newman, 2002), the Louvain algorithm (Blondel et al., 2008), overlapping stochastic block models (Latouche et al., 2011), and the recent biologically-driven algorithm manta (Röttjers & Faust, 2020).
- Line 253: [network] degree > [node] degree. However, I suggest simply “degree” and adding in Line 252-253: “Degree and closeness centrality were estimated [for each node] with the NetworkAnalyzer plugin. Degree refers to [..], whereas closeness [..]”. Give a short explanation of what these graph theoretic measurements indicate in your biological application (microbial relationships/system).
- Line 255: I suggest using in both cases [node/s] instead of [ASV/s] since it is the method section. Graph-theoretic properties are computed for nodes - these nodes represent ASVs.
- Table 3 shows the number of (positive and negative) edges. In addition, it would be interesting to see the edge density, i.e., how many of all the possible edges were observed? Given 150 Protists and 150 Bacteria, there could be potentially 11175 protist-protist, 11175 bacteria-bacteria, and 22500 protist-bacteria associations. Edge density would show the fraction of realized edges from these potential ones, further characterizing the two clusters: cluster 1 being more connected than cluster 2. Subsequently: What would be the implication for the biological system?
- Line 347: instead of [network], the [node] closeness centrality and degree was determined
- Line 354-355: microbial interactions form/are represented through an interaction network (theoretical concept), which is estimated via an association network (used in this study)
- Line 356-359, and 469: [network connectivity] can be determined via [edge density] and [global clustering] but was not shown in the results. If added, please include definition and biological indication in the methods section (Line 252-261).
- Line 401: Please clarify [network variability] - the true microbial interactions or the inferred association networks?
- Line 419: [network variability] (here the two inferred networks) - having only two networks helps to see differences between the two. However, it is challenging to understand network variability with only two networks.
- Line 451-452: Kellogg et al., 2019, did not separate networks but constructed one network per season.
- Fig S4: In the current visualization, networks appear dense and transmit little information. Consider another visualization algorithm or “zooming out.” I suggest a similar color code for protist vs. bacteria vs. edges. For example: positive edges = black and negative edges = red (or two separate networks showing them separately) + one specific color palette (e.g. brown-orange-yellow) for protist vs. another palette (e.g. violet-blue) for bacteria. Another possibility would be to separate nodes based on microbial taxon group, and color nodes based on node degree (Fig. A) and closeness centrality (Fig. B) with the same color scale for both clusters, which will support the result stated in Line 317-318.

Minor comments
- Line 67: “decades” (plural) rather than decade
- Line 77-78: Supporting studies may include (Barraclough, 2015; Piccardi et al., 2019; Hernandez et al., 2021).
- Line 117-118: I agree that defining groups based on diversity may be better than pre-defining groups within pre-network construction methods. However, the authors may want to tone down the statement or provide supporting references concerning post-network construction methods.
- Line 128-129: I suggest “Identifying and characterizing microbial relationships…”
- Line 210: singletons were removed for downstream analysis vs. Line 227: used for diversity estimates. Suggestion: add “unless specified otherwise” to Line 210
- Line 236 “relative count tables” > [relative] vs [count]. Relative frequencies are computed from count tables
- Line 240-241: SPIEC-EASI [is] robust [..] and [results] in sparse networks > [aims to be] robust [..] and [aims to infer] sparse networks. Inferred networks may still be dense.
- Line 249: center log [normalization] of ASV count tables > centered log [ratio (clr) transformation]
- Line 259: [estimated] > I suggest “number of positive and negative edges were [compared] between networks.”
- Line 383: consider rewriting/structuring, [we]?
- Line 392: I was confused by [or …]. Adding [or when the] or restructuring the sentence may help readability
- In several cases, [and] may have been the better choice over [or], e.g., in lines 226, 401, 402, 441, 446
- Some tools should be referenced in the method section. For example, in Line 207: ranacapa: (Kandlikar & Cowen, 2018), Lines 211-212: citation and relevant R-package for dbRDA, Line 223: local regression (loess) curves, Line 227: phyloseq, Line 253: NetworkAnalyzer plugin: (Assenov et al., 2008)
- What has been used to determine the average (mean or median)? Please specify in the text in Lines: 204, 205, 230, 279, 288, for Line 309 the average indicated the mean as stated in Table 2 description - I suggest, stating it also here “Mean observed [..]”
- Line 359-362: Convoluted sentence, which is hard to understand. Please clarify. This statement is also confusing because the protist associations are almost half of all associations in cluster 1 (Table 3).
- Line 403: [seasonal] networks – the first and only time the two networks were termed “seasonal.” The term may not be correct given that the network for cluster 1 contained fall+winter+spring months.
- Line 441: indirect [or] false positive edges > Indirect edges, i.e., a link that has been predicted due to an indirect dependency between two microorganisms via a third factor, are false-positive inferred interactions. However, not every inferred edge that is not a true interaction, i.e., false positive, is due to an underlying indirect dependency relationship.
- Line 444: “reduce ambiguous correlations”: EnDED is specifically designed to reduce environmentally-driven associations (=indirect dependency due to environmental factors) after network construction. SPIEC-EASI aims to account for indirect dependencies during network construction between microbes and does not consider environmental data. However, the recent update of SPIEC-EASI allows accounting for hidden factors (Kurtz et al., 2019).
- Line 446: verified[ ]by > verified[, e.g., ]by. Microbial interaction hypotheses are [verified] by experimental testing, and these may have already been recorded in the primary literature and interaction databases.
- Line 462-464: I agree with the sentence that such correlation analysis could not be done in the present study using two networks. Previous studies aimed to do so, e.g., for spatial data (Chaffron et al., 2021).
- Figures 1 and 2: I like the Figures. The x-axis labels overlap slightly.
- Figure 3: beautiful color scale – but some months are hard to distinguish
- Figure S3: Suggestion: consider rotating the Figure by 90 degrees, reducing the height of the dendrogram, and widening the width for better readability of the months. Date IDs may remain, but colored node leaves (color code from Fig 3) could be used for easier identification. Consider indicating clusters 1 and 2 in all figures (they do not appear in S3).
- Fig 4, S5, S6: I suggest switching the two figures to align with the text (Line 252-255 and 317) describing degree before closeness
- Table S2: Data is sorted based on month, which is ok, but sorting them by actual date would align with the x-axis in Fig 1 and 2. Not needed for a revision, but it may be helpful to add a column for the date (month), introduce a color gradient for the environmental factors (to identify peaks quickly), and indicate clusters via color or boxes.


ASSENOV, Y., RAMÍREZ, F., SCHELHORN, S.-E., LENGAUER, T., & ALBRECHT, M. (2008) Computing topological parameters of biological networks. Bioinformatics, 24, 282–284.

BARRACLOUGH, T.G. (2015) How Do Species Interactions Affect Evolutionary Dynamics Across Whole Communities? Annu. Rev. Ecol. Evol. Syst., 46, 25–48.

BLONDEL, V.D., GUILLAUME, J.-L., LAMBIOTTE, R., & LEFEBVRE, E. (2008) Fast unfolding of communities in large networks. Journal of Statistical Mechanics: Theory and Experiment, 2008, P10008.

CHAFFRON, S., DELAGE, E., BUDINICH, M., VINTACHE, D., HENRY, N., NEF, C., ARDYNA, M., ZAYED, A.A., JUNGER, P.C., GALAND, P.E., LOVEJOY, C., MURRAY, A.E., SARMENTO, H., ACINAS, S.G., BABIN, M., IUDICONE, D., JAILLON, O., KARSENTI, E., WINCKER, P., KARP-BOSS, L., SULLIVAN, M.B., BOWLER, C., DE VARGAS, C., & EVEILLARD, D. (2021) Environmental vulnerability of the global ocean epipelagic plankton community interactome. Sci Adv, 7.

GIRVAN, M. & NEWMAN, M.E.J. (2002) Community structure in social and biological networks. Proc Natl Acad Sci USA, 99, 7821.

HERNANDEZ, D.J., DAVID, A.S., MENGES, E.S., SEARCY, C.A., & AFKHAMI, M.E. (2021) Environmental stress destabilizes microbial networks. The ISME Journal, DOI: 10.1038/s41396-020-00882-x.

KANDLIKAR, G. & COWEN, M. (2018) gauravsk/ranacapa: First release of ranacapa. Zenodo.

KURTZ, Z.D., BONNEAU, R., & MÜLLER, C.L. (2019) Disentangling microbial associations from hidden environmental and technical factors via latent graphical models. bioRxiv, DOI: 10.1101/2019.12.21.885889.

LATOUCHE, P., BIRMELÉ, E., & AMBROISE, C. (2011) Overlapping stochastic block models with application to the French political blogosphere. The Annals of Applied Statistics, 5, 309–336.

PICCARDI, P., VESSMAN, B., & MITRI, S. (2019) Toxicity drives facilitation between 4 bacterial species. Proc Natl Acad Sci USA, 116, 15979.

RÖTTJERS, L. & FAUST, K. (2020) manta: a Clustering Algorithm for Weighted Ecological Networks. mSystems, 5.

---

## Round 0.2 · Minor Revisions

I agree with the reviweers that minor revisions are still necessary. Please make sure to use page and/or line numbers when referring to specific modifications when replying to the edits made.

·

Basic reporting

no comment

Experimental design

Thank you for the revisions and for providing additional details in the github repository for this study.

A couple more suggested edits:

Line 214, write "packages INCLUDING...", rather than "packages like..."

Line 215 (and lines 47, 74, 140, 178 etc...), My comment regarding protists, and the data used to represent protists, was I think slightly misunderstood. I was looking for more precision in identifying the filtering parameters used for the 18S data. So this phrase is more precise about the data used for the remaining analysis, thank you: "To focus on single-celled eukaryotes, we removed Metazoa and Streptophyta that were amplified with the 18S primers." However, there are lineages of eukaryotes that are not in the Metazoa and Streptophyta, such as brown algae (Phaeophyta, Stramenopiles), that are multicellular. So you should stick to using the term protists, and not single-celled eukaryotes for eukaryotes that are not in the Metazoa and Streptophyta. Arguably, Fungi are not protists as well. But if you left these in, then I think the term protists is still better than single-celled eukaryotes.


Lines 263-264 - In Line 345, Table 2 and Table S3, there seems to be only 297 ASVs in the cluster 2 network analysis, not 300. Please explain or clarify.

Validity of the findings

Thank you for clarifying the dbRDA analysis and providing a more detailed description of the network results. I just have a few more comments on the network analysis to address:

Lines 347-348, Table S3 - of the 597 ASVs in Table S3, they are labeled as either belonging to Cluster 1 or Cluster 2. So does that mean that there were no ASVs (which are the most abundant ASVs) that were shared or overlapping between the clusters? If that is the case, then the nodes were completely different, not "often different". But that is very surprising, that there were no overlapping ASVs for the top 150 most abundant ASVs between the two clusters. If there are shared ASVs, they should be indicated as such in Table S3. The shared ASVs can be listed in the middle, between the list of ASVs that are in cluster 1 only and cluster 2 only.

Line 348 - The read counts of ASVs (nodes) should be added to Table S3 to support the statement "Nodes present only in Cluster 1 or 2 networks were represented by less abundant groups". However, based on Table S3, the data is not presented in a way to easily evaluate if there are any shared ASVs between clusters 1 and 2 (see above). If it is the case that all ASVs are either in Cluster 1 or 2 but not both, then it doesn't make sense that these nodes could be less abundant than shared nodes because there aren't any shared nodes.

Lines 357-358. For the most part, Cluster 1 network statistics are only slightly higher than Cluster 2. So, should modify the result to say: "Edge density was SLIGHTLY higher (and average path length lower) in Cluster 1 for the overall network or when networks were analyzed for each domain level pairing (Table 2).", rather than "typically" higher. The biological significance of this increase is likely small.

Lines 351-354, 366-377, 384-388 (also throughout)- The overall description of the network analysis is greatly improved and clarified, especially in lines 384-388 where the information that can be interpreted from co-occurrence networks is discussed. The authors should also be clear and state earlier in the Results that in describing the edges as associations or relationships, that these are statistical associations, more specifically correlations, that may indicate biological associations or similar environmental responses - but even so, may not be direct associations, relationships or interactions. I think it is important to first define the edges as correlations or statistically significant associations in abundance that may potentially indicate biological interactions (direct or indirect), or similarity in responses to environmental factors. And be clear in distinguishing "relationships" as edges, and "relationships" as biological relationships.

for eg. Lines 411-412 could be changed to: Prokaryote-Prokaryote and Prokaryote-Eukaryote EDGES increased IN NUMBER by >100% in Cluster 1, indicating these types of relationships may have been preferable in the estuary...

Lines 507-511 - This phrase: "may limit detection of more rare microbial relationships" is not quite accurate. By building networks with the most abundant taxa, this will limit the detection of correlations with and among taxa of lower abundance, but these putative associations may not be rare in the community. i.e. these taxa may be common in the community and form consistent associations (potentially with abundant taxa), but the taxa do not occur in high abundance.

Additional comments

Minor edits:

Lines 293-294 - For temperature correlating with NO3 and PO4, there should be two correlation values reported, one for Temp v. NO3 and one for Temp v. PO4, but only one R and p-value are provided.

Lines 328-330, Fig. 3 - The percentage variation explained by each axis should be indicated in the dbRDA plots.

Line 362 - I believe this result should refer to Table S3, not S4.

Lines 366-378 - Mention that the comparisons described in this paragraph are focusing on the positive correlations.

Figure 1B - For this phrase: "Only significant correlations are shown", be more accurate and state that significant correlations are coloured, whereas white boxes indicate no significant correlation.

Legends for supplemental figures and tables. From the available revised files, the legends were still not showing up with the supplemental figures and tables. For Supplemental Figures and Tables, I would include the legend as part of the Figure or Table (e.g. add a row that includes the legend at the top of the excel spreadsheets, add in a text box in the figure for the legend). Or confirm with the journal how the legends for the supplementary material should be included.

·

Basic reporting

I thank Sean R. Anderson and Elizabeth L. Harvey for addressing my comments, giving clarifications, and correcting network-related descriptions. For example:
- Selection of 300 ASVs for both clusters
- Archaea ASVs
- Filtering strategy
- Figure 4, several Supplementary Figures, Table 2 (former 3), and Supplementary Table S2
- The rewritten discussion part (497-511) reads well, states differences between methods, and mentions possible drawbacks for rare groups.
- Missing files were added to the GitHub repository, and authors provided archived files on Zenodo.
- Network terminology is appropriately used (see little minor comments below).

Moreover, the fraction of shared taxa (order level) and "network Comp"-sheet specifying shared and unique taxa and their number of nodes are valuable to the analysis.

Experimental design

no comment

Validity of the findings

no comment

Additional comments

Minor comments:
1) l. 84: "Microbes are also driven by [interactions they] have with each other" - "[interactions that they]"
2) l.121-124: "Networks can also be separated with clustering techniques, binning nodes into multiple clusters (Röttjers & Faust, 2018; Chaffron et al., 2021); however, network clustering tools are limited (Faust, 2021) and have not been well applied to marine samples." - Suggestion to avoid the term "network clustering": "Networks can also be separated by binning nodes (Röttjers & Faust, 2018; Chaffron et al., 2021) but tools are limited (Faust, 2021) and have not been well applied to marine samples."
3) l 227: "were rarefied to the minimum read count for 16S and 18S tables, respectively" - specific number: "minimum read count of XX"
4) l.271-272: "centered log ratio (clr) [normalization]" - as suggested in previous review and also stated in paper referenced (Tipton et al., 2018) by authors: "[normalization]" should be substituted for "[transformation]". Also see (Tipton et al., 2018) between equation (1) and (2): "centered log-ratio (CLR) [transformation]"
5) l. 397 Discussion "node centrality, degree, and edge number" vs. conclusion l. 535-536 "network centrality, degree, and edge number" - network > node
6) l. 399: "being represented by similar order level groups." - A bit more than half of the taxa are in common. I suggest rewriting or "They are being mainly (56%) represented by similar..."
7) l 529: "spatial time points (Chaffron et al. 2021)." - spatial samples
8) Authors comment: "Average was based on the mean. We have revised throughout the text. " - not done in l. 235, 243, 281, 316, 323, 495, 509. It may be sufficient to state once (mean is used when referring to average throughout the text)

Very minor comment:
9) Thank you for addressing the plenitude of my comments and giving clarifications if needed. Still, it would have helped me a lot if line numbers had been given in addition to "revised/clarified" ;)

---

## Round 0.3 · accepted · Accept

Thank you for your conscientious attention to all of the reviewer comments.

·

Basic reporting

no comment

Experimental design

no comment

Validity of the findings

no comment

Additional comments

Thank you for considering all of my comments, which all have been thoughtfully and properly addressed. The care that you have taken in revising this manuscript is much appreciated.

·

Basic reporting

Congratulations, Sean R. Anderson and Elizabeth L. Harvey. The manuscript reads nicely - I especially enjoyed the network descriptions and interpretations.

Experimental design

-

Validity of the findings

-

Additional comments

l. 98 in cleaned version: “Amplicon surveys coupled with co-occurrence network analysis [represent]” (instead of “represents”??)